# Light chain 2 is a Tctex-type related axonemal dynein light chain that regulates directional ciliary motility in *Trypanosoma brucei*

**Subash Godar**[1,2¤a], **James Oristian**[2,3¤b], **Valerie Hinsch**[2,3¤c], **Katherine Wentworth**[2,4¤d], **Ethan Lopez**[2,3], **Parastoo Amlashi**[2,4¤e], **Gerald Enverso**[2,4¤b], **Samantha Markley**[2,4], **Joshua Daniel Alper**[1,2,4¤f] *

**1** Department of Physics and Astronomy, College of Science, Clemson University, Clemson, South Carolina, United States of America, **2** Eukaryotic Pathogens Innovation Center, Clemson University, Clemson, South Carolina, United States of America, **3** Department of Genetics and Biochemistry, College of Science, Clemson University, Clemson, South Carolina, United States of America, **4** Department of Biological Sciences, College of Science, Clemson University, Clemson, South Carolina, United States of America

¤a Current address: Shared Equipment Authority, Rice University, Houston, Texas, United States of America
¤b Current address: Department of Infectious Disease and the Center for Tropical and Emerging Global Disease, University of Georgia, Athens, Georgia, United States of America
¤c Current address: Department of Biological Sciences, University of South Carolina, Columbia, South Carolina, United States of America
¤d Current address: Department of Molecular, Cellular, and Developmental Biology, University of Michigan, Ann Arbor, Michigan, United States of America
¤e Current address: Koch Institute of Integrative Cancer Research, Massachusetts Institute of Technology, Cambridge, Massachusetts, United States of America
¤f Current address: GlaxoSmithKline, Cambridge, Massachusetts, United States of America
* alper@clemson.edu

**Data Availability Statement:** All relevant data are within the manuscript and its Supporting information files.

## Abstract

Flagellar motility is essential for the cell morphology, viability, and virulence of pathogenic kinetoplastids. *Trypanosoma brucei* flagella beat with a bending wave that propagates from the flagellum's tip to its base, rather than base-to-tip as in other eukaryotes. Thousands of dynein motor proteins coordinate their activity to drive ciliary bending wave propagation. Dynein-associated light and intermediate chains regulate the biophysical mechanisms of axonemal dynein. Tctex-type outer arm dynein light chain 2 (LC2) regulates flagellar bending wave propagation direction, amplitude, and frequency in *Chlamydomonas reinhardtii*. However, the role of Tctex-type light chains in regulating *T. brucei* motility is unknown. Here, we used a combination of bioinformatics, in-situ molecular tagging, and immunofluorescence microscopy to identify a Tctex-type light chain in the procyclic form of *T. brucei* (TbLC2). We knocked down TbLC2 expression using RNAi in both wild-type and FLAM3, a flagellar attachment zone protein, knockdown cells and quantified TbLC2's effects on trypanosome cell biology and biophysics. We found that TbLC2 knockdown reduced the directional persistence of trypanosome cell swimming, induced an asymmetric ciliary bending waveform, modulated the bias between the base-to-tip and tip-to-base beating modes, and increased the beating frequency. Together, our findings are consistent with a model of TbLC2 as a down-regulator of axonemal dynein activity that stabilizes the forward tip-to-base beating ciliary waveform characteristic of trypanosome cells. Our work sheds light on axonemal dynein regulation mechanisms that contribute to pathogenic kinetoplastids'

**Funding:** This work was supported by the National Institute of Allergy and Infectious Diseases (NIAID) of the National Institutes of Health under award number R15AI137979 (JDA) and the National Institute of General Medical Sciences (NIGMS) of the National Institutes of Health under award number P20GM109094 (JDA). Additional funding was provided by the Clemson University Division of Research, College of Science, Department of Physics and Astronomy, Creative Inquiry program, and University Professional Internship and Co-op Program (JDA). The funders had no role in study design, data collection and analysis, decision to publish, or preparation of the manuscript. URLs: https://www.niaid.nih.gov/, https://www.nigms.nih. gov/, https://www.clemson.edu/.

**Competing interests:** The authors have declared that no competing interests exist.

unique tip-to-base ciliary beating nature and how those mechanisms underlie dynein-driven ciliary motility more generally.

## Author summary

Kinetoplastea is a class of ciliated protists that include parasitic trypanosomes, which cause severe disease in people and livestock in tropical regions across the globe. All trypanosomes, including *Trypanosoma brucei*, require a cilium to provide propulsive force for directional swimming motility, host immune system evasion, and various aspects of their cell cycle. Thus, a functional cilium is essential for the virulence of the parasite.

Trypanosome cilia exhibit a unique tip-to-base beating mechanism, different from the base-to-tip beating of most other eukaryotic cilia. Multiple ciliary proteins are involved in the complex biophysical and biochemical mechanisms that underlie the trypanosome ciliary beating. These include dynein motor proteins that power the beat, dynein-related light chains that regulate the beat, and many other proteins in the nexin-dynein regulatory complex, in the radial spokes, and associated with the central pair of microtubules, for example.

Here, we identify a Tctex-type dynein light chain in *T. brucei* that we named TbLC2 because it has sequence homology, structural similarity, and ciliary localization like LC2 homologs in other organisms. We demonstrate that TbLC2 has dynein regulatory functions, with implications on the unique aspects of trypanosome ciliary beating and cellular swimming motility. Our study represents an additional step toward understanding the functions of the trypanosome ciliary proteome, which could provide novel therapeutic targets against the unique aspects of trypanosome ciliary motility.

## Introduction

Motile cilia and flagella (terms used interchangeably, here we favor cilia) are multifunctional and dynamic eukaryotic cell organelles that drive cell motility, generate fluid flows across surfaces, sense mechanical and chemical signals, and serve as cellular adhesion and secretion sites [1]. *Trypanosoma brucei*, a pathogenic Kinetoplastea class member that causes Human African Trypanosomiasis (HAT) and nagana in livestock, has a single motile cilium attached to its cell body by the flagellar attachment zone (FAZ) [2]. Ciliary motility is essential throughout the life cycle of the parasite [3]. Trypanosomes use their cilia to generate propulsive force for directional swimming, proper morphogenesis, completion of cytokinesis, and initiation of kinetoplast (a network of interlocked circular DNA, kDNA, inside a large, single mitochondrion associated with the basal body [4]) segregation during cell division [5]. The cilium plays a significant role in tsetse fly host midgut epithelial layer attachment, and persistent swimming motility is essential for migration from the tsetse fly's midgut to the salivary gland, social motility [6, 7], chemotaxis [8], and evasion of the immune system within the highly viscous fluid and tissue environment of the trypanosome's mammalian host [9]. Therefore, proteins enabling and regulating cilium-driven swimming motility are vital for parasite viability and pathogenicity [10], and they are a significant virulence factor [3, 10, 11]. Through the combination of proteomic [12] and molecular mechanistic studies, it is thought that ciliary proteins essential for motility could provide an excellent prospect for novel therapeutic drug targets [3].

Trypanosome cilia generate motile force using their axoneme, a highly-conserved structure made of a "9+2" arrangement of microtubule doublets, dynein motor proteins, and

approximately 750 other ciliary proteins [12]. Inner (IAD) and outer arm dyneins (OAD) [13] are microtubule minus-end-directed motor protein complexes [13] comprising multiple heavy chains (HCs), intermediate chains (ICs), and light chains (LCs) that power ciliary motility. Dynein heavy chain tail domains attach dynein to one microtubule doublet through the docking complex [14] while its microtubule-binding domain (MTBD) binds to the adjacent doublet [15] and slides it distally in an ATP-dependent manner [16, 17]. The nexin-dynein regulatory complex (N-DRC) and basal constraints convert doublet sliding to bending [18]. Coordination mechanisms that activate [19] and deactivate [20] dynein along and across the axoneme ultimately orchestrate a bending wave that is necessary for motile functionality [21]. In *Trypanosoma brucei* and other Trypanosomatidae cells, this bending wave propagates from the tip to the base of the cilium [3], which is different from the canonical base-to-tip bending wave propagation direction found in nearly all other eukaryotic cilia and flagella.

Wild-type *T. brucei* cells exhibit highly directionally persistent swimming motility due to the tip-to-base propagation of their ciliary beat waveform for extended periods [11]. However, occasional pauses followed by a tumbling maneuver transiently interrupts their directionally persistent swimming motility mode [11]. An abrupt switch from the highly symmetric tip-to-base beating waveform to an asymmetric base-to-tip ciliary beating waveform [22, 23] initiates tumbling [2] and reorients the swimming direction. While less than 25% of the wild-type population demonstrate the tumbling mode [11, 24], trypanosome cells with mutations to the N-DRC tumble rather continuously [5, 25], suggesting that dynein motor regulation underlies tip-to-base ciliary beating in the directionally persistent swimming motility mode of trypanosome cells and the switch to the base-to-tip driven tumbling motility mode.

There is significant additional evidence from multiple species supporting the model of axonemal dynein motors requiring significant regulation [16] to achieve the precise and regular tip-to-base ciliary beat waveforms necessary for persistent directional motility [26]. Inner and outer arm axonemal dynein regulation acts through multiple direct and indirect interactions, including through their ICs and LCs, which respond to various stimuli, including $Ca^{2+}$, phosphorylation, and redox poise [27–29], and in conjunction with the N-DRC, radial spokes, and the central pair complex [17, 18]. For example, LC4, a calmodulin-like outer arm dynein light chain also known as DLE1 [19], induces a $Ca^{2+}$ dependent conformational change in the N-terminal tail domain of the γ HC (DHC15 [19]) that activates the motor in *Chlamydomonas reinhardtii* cells (a single-celled alga that is commonly used as a model organism for ciliary motility) [28]. The loss of an LC4-like protein from *Leishmania mexicana*, another pathogenic Kinetoplastea, results in a higher ciliary beat frequency and an increased swimming speed [25]. LC1, a leucine-rich repeat dynein light chain, directly interacts with the MTBD of γ HC in *Chlamydomonas* cells [20]. The loss of LC1 from *C. reinhardtii* and *T. brucei* changes the ciliary beat frequency and beating mode, which reverses the direction of motility [30, 31].

LC2 is a member of the Tctex-type family, and it is essential for OAD assembly in *Chlamydomonas* [32]. However, *Chlamydomonas* cells with an N-terminal truncation on LC2 have partially functional cilia with the OADs still intact [15], demonstrating that mutations affecting the dynein regulatory function of LC2 can cause ciliary motility defects separate from the structural defects caused by the complete loss of the protein. *Chlamydomonas* LC2 has no direct interactions with the dynein HCs during any stage of their mechanochemical cycle [26], yet it exhibits OAD regulatory function [15], which suggests LC-LC or LC-IC interactions may be necessary for LC2 to carry out its regulatory function. Additionally, inactivation of Tcte3-3, an LC2 homolog in mice, causes motility defects in sperm cells characterized by reduced motility persistence [27]. Inactivation of Tcte3-3 also increased the ciliary waveform's beat frequency and decreased its amplitude in spermatozoa, rendering them unable to migrate into

the female oviduct [27]. Similar to the *Chlamydomonas* mutant with LC2 N-terminus truncation [15], murine cilia missing Tcte3-3 showed no structural defects within the axoneme, including intact OADs [27], but some spermatozoa had two cilia, and others showed a bent or coiled phenotype [27]. Moreover, LC2 is phosphorylated upon the activation of sperm motility in rainbow trout and chum salmon [28], and reduced phosphorylation of LC2 correlates with less progressive swimming of the spermatozoa [28]. In total, these results suggest that LC2 is an important regulator of axonemal dynein function and show its loss or mutation leads to ciliary motility defects. However, a quantitative functional analysis of the roles played by LC2 in regulating the ciliary motility, cell morphology, and ultimately in the swimming behavior of Kinetoplastea, including disease-causing parasites like *T. brucei*, is lacking.

In this study, we identified a trypanosome homolog of the *Chlamydomonas* Tctex1/Tctex2 family outer arm dynein light chain 2 (TbLC2). We knocked down the expression of endogenous TbLC2 in *T. brucei* cells using RNA interference and characterized the effects of TbLC2 knockdown on multiple cell biological and trypanosome cell swimming motility characteristics. In doing so, we elucidated some mechanisms by which TbLC2 regulates axonemal dynein.

## Results

### TbLC2 shows high sequence and structural conservation with outer arm dynein Tctex-type light chains

We searched for Tctex-type dynein light chains in the *Trypanosoma brucei* genome (TriTrypDB [33]) using the *C. reinhardtii* outer arm axonemal dynein light chain 2 (Cre12. g527750 in *C. reinhardtii* v5.6 [29, 34], which we refer to as CrLC2) as the query sequence in protein BLAST [35]. We identified Tb927.9.12820 and Tb927.11.7740 as two putative Tctextype dynein light chain homologs (Materials and methods [36]). We cross-referenced these two hits for ciliary localization in the TrypTag database [37], and we found that only Tb927.9.12820, previously annotated as a putative dynein light chain [33], strongly localized to the cilium. Therefore, hereafter we refer to Tb927.9.12820 as *Trypanosoma brucei* dynein light chain 2 (TbLC2).

We aligned TbLC2 with LC2 proteins from multiple species (protozoans to humans, S1 Fig). We found 29% sequence identity over a 124-amino acid region of overlap between *C. reinhardtii* and *T. brucei* (S1 Fig). We also identified significant extensions in the N-terminal domain of mouse (*Mus musculus*) and rainbow trout (*Oncorhynchus mykiss*) LC2, as well as gaps in the N-terminal regions of *T. brucei* and *Trypanosoma cruzi* LC2 (S1 Fig).

We generated sequence-based structural homology and template-free models of TbLC2 using SWISS-MODEL (S1 File) [38] and AlphaFold (S2 and S3 Files) [39] (Materials and methods), respectively. We compared the cryo-EM structure of CrLC2 in the outer arm dynein core subcomplex from *C. reinhardtii* (PDB ID: 7KZN [26]) to these models (Fig 1A). We found a high degree of structural alignment between CrLC2 and our homology and templatefree models of TbLC2 (Fig 1A, RMSD = 0.1 Å and 1.7 Å, respectively). We also found a high degree of local surface charge density conservation between CrLC2 and the models of TbLC2 (Fig 1B–1D, lower panels), particularly in the suggested binding interface between CrLC2 and CrLC9 (S2 Fig) [26]. Of the 35 residues in CrLC2 constituting the suggested binding interface (S2 Fig), 12 residues were identical and 20 were hydrophobicity and polarity preserving substitutions (S2 Fig). In particular, R100 and R104 in CrLC2, which constitute 3/9 of the polar contacts with the adjacent CrLC9, were conserved (identical, R88, and charge conserving, K92, respectively) in TbLC2 (Figs 1A and S2).

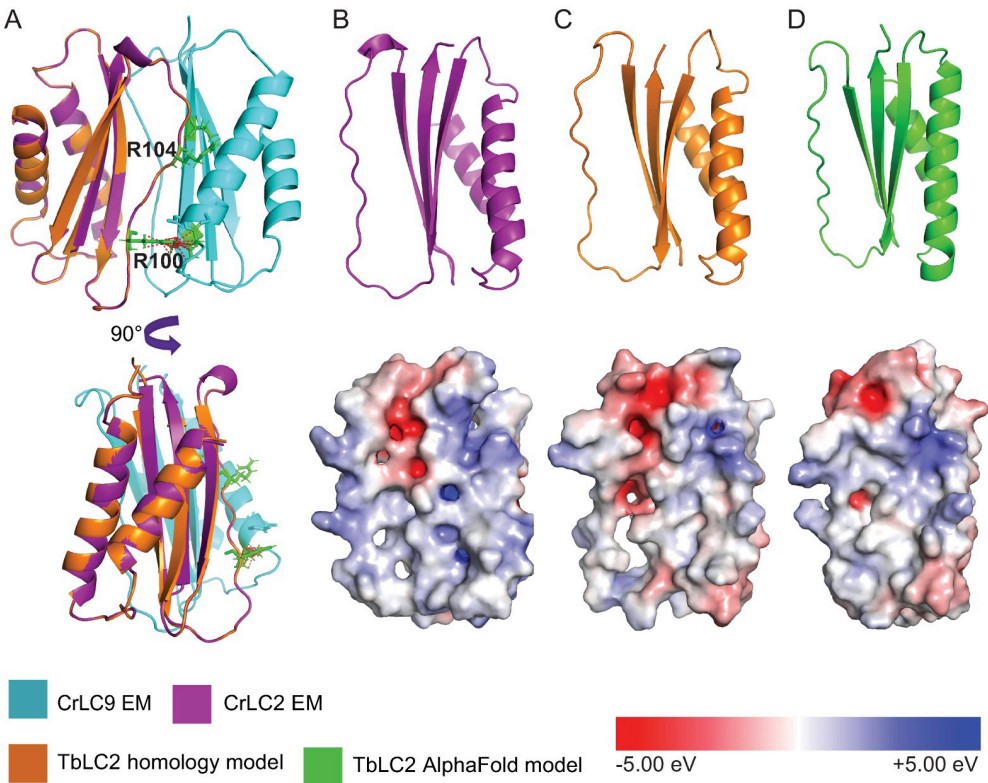

**Fig 1. TbLC2 shows a high degree of structural and electrical potential conservation with the *C. reinhardtii* LC2 protein. A.** Alignment of the sequence-based structural homology model of TbLC2 (*orange*) with LC2 and LC9 (denoted as CrLC2, *purple*, and CrLC9, *cyan*, respectively) from the *C. reinhardtii* outer dynein arm core subcomplex structure (PDB ID: 7KZN [26]). R100 and R104 (*green*) in CrLC2, which are highly conserved polar residues, have extensive contacts (*red dotted line*) with CrLC9. The structures (*top*) of CrLC2 in **B.**, the structural homology model of TbLC2 in **C.**, and the template-free model of TbLC2 in **D.**, are shown with surface charge density mapped to the electrostatic potential distribution (*bottom*). The homology model of TbLC2 in panels A and C (*orange*) was generated by SWISS MODEL and used and modified under the creative commons license CC BY-SA 4.0.

## TbLC2 localizes along the axoneme

To examine the cellular localization of TbLC2 in trypanosome cells, we overexpressed recombinant, eGFP-tagged LC2 (TbLC2::eGFP) in wild-type procyclic-form trypanosome cells using the tetracycline-inducible pLEW100V5 expression vector system (hereafter called WT/LC2 OE cells, Table B in S1 Text for *T. brucei* strains generated and used in this study, Material and methods). We verified the expression of the TbLC2::eGFP in WT/LC2 OE cells using flow cytometry (Material and methods) and found that WT/LC2 OE cells had approximately 100-fold higher eGFP fluorescence intensity than the background fluorescence observed in the parental wild-type cells (S3 Fig). We also imaged the WT/LC2 OE cells (Material and methods) and found high levels of eGFP fluorescence throughout the cells (Fig 2A). To confirm ciliary localization of TbLC2, we immunostained the cells (Materials and methods) using a primary antibody against paraflagellar rod 2 protein (PFR2 [10]). We observed TbLC2::eGFP (Fig 2A, *green*) and PFR2 (Fig 2A, *red*) signals along most of the cilium's length, apart from the tip where the PFR2 signal ran shorter than the eGFP signal (Fig 2A) suggesting the ciliary localization of TbLC2. However, the high cytoplasmic fluorescence signal prevented us from conclusively confirming the localization of TbLC2 to the cilium because it was attached to the cell body (Fig 2A).

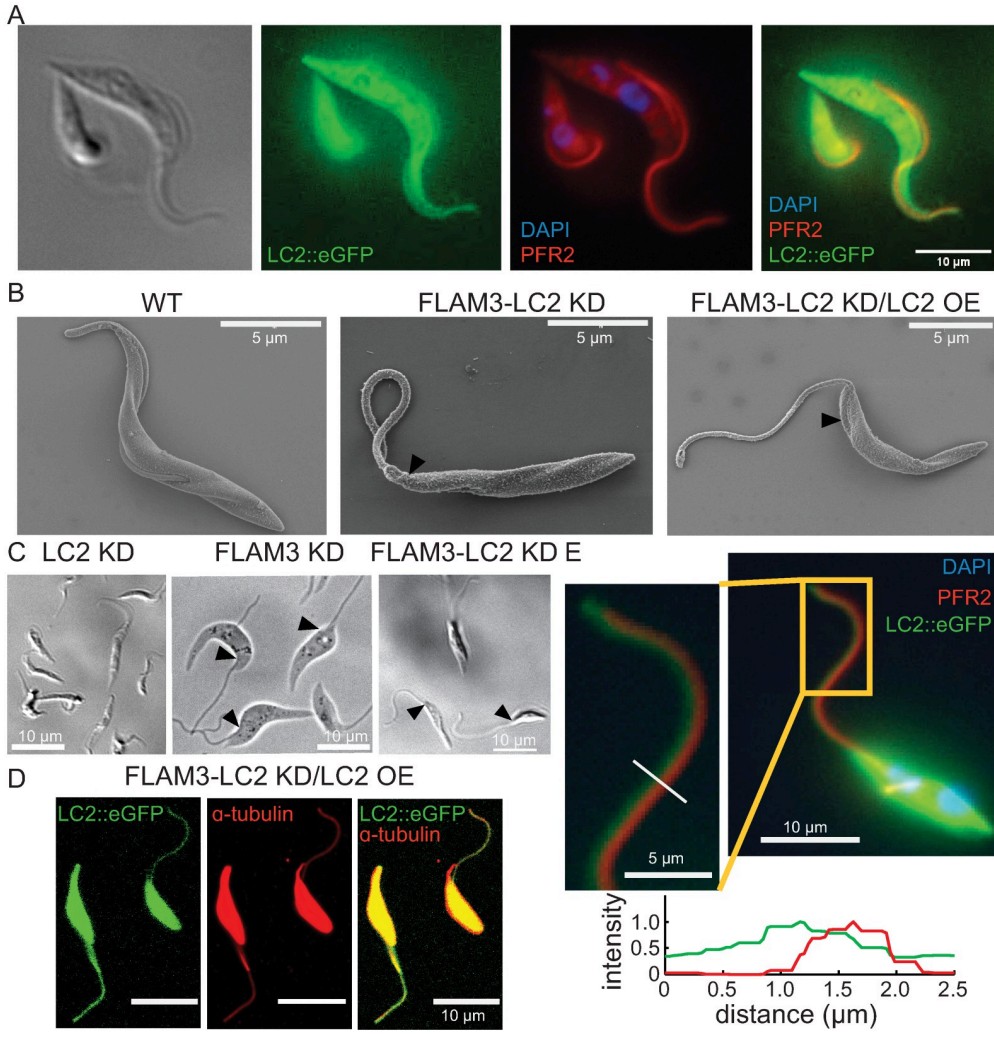

**Fig 2. Overexpressed TbLC2 localized to the axoneme as well as the cytoplasm. A**. Doxycycline induced TbLC2::eGFP overexpressing wild-type cells (WT/LC2 OE cell) visualized using differential interference contrast (DIC) microscopy (*left*) and under widefield fluorescence microscopy (*right*). The TbLC2::eGFP (*green*) localized to the cytoplasm and the cilium. The immunostained paraflagellar rod 2 (PFR2, *red*), DAPI stained DNA (DAPI, *blue*), and TbLC2::eGFP overlay (*right*), suggest that the TbLC2 has ciliary and cytoplasmic localization. **B**. SEM images showing cilium and cell body morphology of wild-type (WT, *left*), FLAM3-LC2 KD (*middle*), and FLAM3-LC2 KD/LC2 OE (*right*) cells showing short flagellar attachment zones (*arrows*) and nearly complete ciliary detachment from the cell body in the FLAM3 knocked down cells. The scale bar is 5 μm in each panel. **C**. LC2 KD (*left*), FLAM3 KD (*center*), and FLAM3-LC2 KD (*right*) cells visualized with DIC microscopy showing short flagellar attachment zones (*arrows*) and nearly complete ciliary detachment from the cell body. **D**. TbLC2::eGFP (*green*) expressing FLAM3-LC2 KD/LC2 OE cells stained with anti-α-tubulin antibody (*red*) visualized with a confocal microscope. **E**. TbLC2::eGFP (*green*) expressing FLAM3-LC2 KD/LC2 OE cells stained with DAPI (*blue*) and the anti-PFR2 antibody (*red*) visualized under a widefield fluorescence microscope. The inset shows that TbLC2::eGFP and the paraflagellar rod are separately localized, and the line scan plots shows the normalized fluorescence intensity of TbLC2::eGFP (*green*) and PFR2 (*red*) along the line (*white*) shown in the inset.

We used RNA interference to knock down FLAM3, a FAZ protein essential for maintaining the cilium's attachment along the cell body (FLAM3 KD cells, Materials and methods [40]), to more clearly distinguish LC2::eGFP in the cilium from the eGFP signal in the cell body. We also knocked down endogenous TbLC2 expression using RNAi in wild-type (hereafter called LC2 KD cells, Materials and methods) and FLAM3 KD cells (hereafter called FLAM3-LC2 KD

cells, Materials and methods) to optimize the efficiency of TbLC2::eGFP assembly into the cilium (hereafter called FLAM3-LC2 KD/LC2 OE cells, Materials and methods). We verified the knockdowns using reverse transcription quantitative PCR (RT-qPCR, Materials and methods [41, 42]). We found a significant reduction of the endogenous LC2 gene mRNA levels in both LC2 KD and FLAM3-LC2 KD cells (reduced to 52.9 ± 2.9% and 43.4 ± 3.8% of the wild-type level, mean ± standard error of the mean) and the FLAM3 gene mRNA level in FLAM3-LC2 KD cells (16.1 ± 3.7% of the wild-type level) after 72 hrs of induction (S4 Fig).

As previously described for FLAM3 knockdown in procyclic form trypanosomes [40], we found that FLAM3 knockdown in both wild-type and LC2 KD cells significantly reduced the FAZ length (arrows in Fig 2B and 2C), as compared to LC2 KD cells (Fig 2C, *left*), leading to a mostly detached cilium held onto the cell body only at its proximal end. With the cilium detached from the cell body, we detected eGFP fluorescence along the cilia of FLAM3-LC2 KD/LC2 OE cells (Fig 2D and 2E).

To further isolate the sub-ciliary localization of TbLC2::eGFP and PFR2, we immunostained FLAM3-LC2 KD/LC2 OE cells using a primary antibodies against α-tubulin and PFR2 (Materials and methods). Additionally, we observed that the FLAM3 knockdown-induced detached cilium retained its attachment to the paraflagellar rod (PFR2, *red*) at a fixed, finite separation of 300–400 nm from the eGFP signal (*green*) along most of the cilium's length (Fig 2E) except at the tip where the eGFP signal extended beyond the PFR2 signal (Fig 2E, inset). Together, the immunostaining observations suggest that TbLC2 localized to the axoneme, rather than the paraflagellar rod, as expected [37].

We further confirmed the subcellular localization of TbLC2 to the cilium using cell fractionation and immunoblotting. In a procedure enabled by the separation of the cilium from the cell body in FLAM3 knockdown cells lines, we sheared the cilia off FLAM3-LC2 KD/LC2 OE cells and separated them into cell body and ciliary fractions (Materials and methods). We then prepared lysates from the whole-cell (WC), cell body only (CB), and ciliary (cilia) fractions and probed them by western blot using an anti-GFP antibody (Materials and methods). We observed strong bands at approximately 50 kDa, which corresponds to the molecular weight of TbLC2::eGFP (Materials and methods), in all the fractions of the induced cells, including the ciliary fraction (S5 Fig). This result confirmed the localization of TbLC2 to the cilium.

## TbLC2 is not essential for stable assembly of outer arm dynein in the axoneme

The knockdown of LC2 causes the failure of outer arm dynein to assemble into the axonemes of *Chlamydomonas* cells [43]. Therefore, we visualized dynein assembly into trypanosome axonemes in ciliary cross-sections with transmission electron microscopy (Materials and methods). FLAM3 knockdown does not prevent the transportation of ciliary dynein complexes into the cilium [44, 45], their incorporation into the axoneme [40], or the localization of LC2::eGFP into the axonemes in FLAM3-LC2 KD/LC2 OE cells (Fig 2D and 2E). Thus, it is unlikely that FLAM3 knockdown affects axoneme ultrastructure, so we used the FLAM3 knockdown background cells to examine the ultrastructural differences of the axoneme caused by TbLC2 knockdown.

We found that TbLC2 knockdown left both inner and outer arm dyneins intact in the axoneme of FLAM3-LC2 KD cells (Fig 3, *middle*), like wild-type (Fig 3, *left*) and FLAM3-LC2 KD/LC2 OE (Fig 3, *right*) cells, suggesting that TbLC2 is not essential for assembly of dynein into the cilium. This observation is unlike the *C. reinhardtii* Tctex-type LC2 outer arm dynein light chain mutant. However, the observation is similar to other Tctex-type dynein light chain

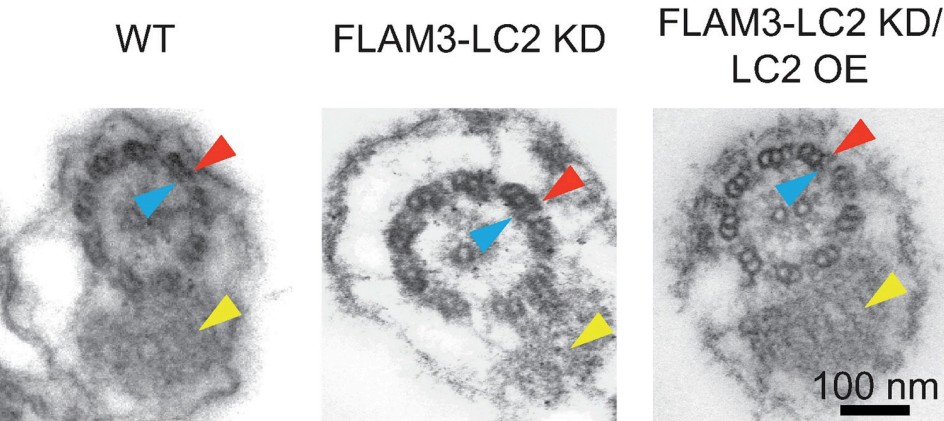

**Fig 3. Stable assembly of outer arm dynein into the axoneme does not require TbLC2.** Transmission electron microscopy (TEM) images of wild-type (WT, *left*), FLAM3-LC2 KD (*middle*), and FLAM3-LC2 KD/LC2 OE (*right*) cells show the axoneme (canonical 9+2 microtubule arrangement) and paraflagellar rod (*yellow arrowheads*). The outer arm (*red arrowheads*) and inner arm (*blue arrowheads*) dyneins were intact in all three cell lines. The scale bar is 100 nm, and all micrographs have the same magnification.

mutant cell lines, including Tcte3 knockout [27], a Tctex2 dynein light chain in mouse outer dynein arm, and Tctex2b knockout [46], a Tctex2-related inner dynein arm light chain in *C. reinhardtii*, which are not necessary for the stable assembly of their respective axonemal dynein arms into the axoneme. Other major structural components of the axoneme, including the central pair and microtubule doublets, also appeared unperturbed by the lack of TbLC2 in the knockdown cells (Fig 3). Additionally, the cilium retained the paraflagellar rod (Fig 3), commensurate with the PFR2 immunofluorescence imaging (Fig 2E). Because we found that the knockdown of TbLC2 did not prevent the assembly of outer arm dyneins into the axoneme (Fig 4) nor change the length of the cilium (S6 Fig) and yet resulted in cell division and kineto-plast localization defects associated with impaired ciliary motility and incomplete cytokinesis,

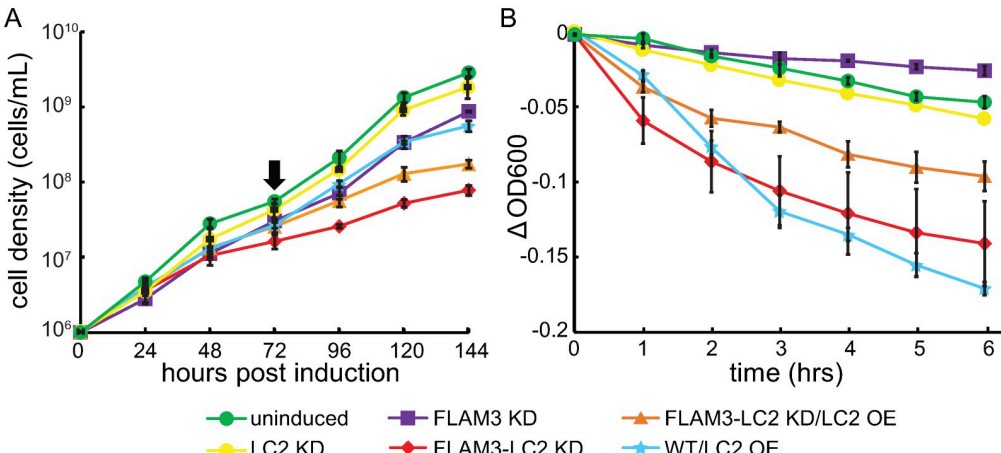

**Fig 4. LC2 knocked down cells show growth and motility defects. A**. Growth curves of various RNAi knocked down cells and uninduced cells (*green*). At 72 hours post-induction (*black arrow*), we diluted the cells back to the starting cell density ($1\times10^6$ cells/mL) and allowed them to grow for another 72 hours. We scaled the cell densities from the diluted culture to reflect the total growth. **B**. Sedimentation curves of various RNAi knocked down cells and uninduced cells (*green*). In both panels, each data point represents the mean from three independent experiments, and the error bars represent the standard error of the mean (SEM). The legend applies to both panels.

particularly in FLAM3 knockdown cells (S7 Fig and S1 Text), these results suggest that TbLC2 has a regulatory role in axonemal dynein molecular mechanisms and the cell biological functions that dynein motility underlies.

## LC2 knockdown results in reduced growth rates

We examined the effect of the RNAi knockdowns on cell growth rate using a cell counting chamber to find the cell density every 24 hours (Materials and methods). We found that LC2 KD cells grew at a comparable rate to uninduced cells with a population doubling time of $12.6 \pm 0.1$ and $12.1 \pm 0.3$ hours, respectively (*p-v*alue = 0.15, two-tailed paired *t*-test, doubling time determined in the first 72 hours of culture, Fig 4A). Additionally, we repeated this analysis in FLAM3 knocked-down cells because FLAM3 knockdown reduces the effective stiffness of the anterior part of the trypanosome's cilium causing the ciliary beating to be highly susceptible to and amplifying the effects of external perturbation and biological noise [47] like genetic perturbations that affect ciliary motility. We found that FLAM3-LC2 KD cells grew at a significantly reduced rate, with a population doubling time of $18.0 \pm 1.4$ hours, as compared to $14.5 \pm 0.2$ hours for FLAM3 KD cells (*p-v*alue = 0.002, two-tailed paired *t*-test, Fig 4A). We also found that FLAM3-LC2 KD cells grew to approximately $1/10^{th}$ and $1/6^{th}$ the density of the uninduced and FLAM3 KD cells after six days post-induction (*p-v*alues < 0.0001 in both cases, two-tailed paired *t*-tests, Fig 4A), respectively, and they showed some cell division phenotypes (S7 Fig and S1 Text).

## LC2 knockdown results in a rapid settling of the cells in solution

To investigate the effect of TbLC2 depletion from wild-type and FLAM3 knockdown cells on their swimming behavior, we performed a sedimentation assay by monitoring the change in absorbance near the top of cell solution at 600 nm over time (Materials and methods). We found that LC2 KD cells settled to the bottom of the cuvette at a slightly faster rate than the uninduced, as determined by the change in OD600 after 6 hours (Fig 4B). We repeated the settling assay in FLAM3 knocked-down cells and found that FLAM3-LC2 KD cells settled to the bottom of the cuvette at a faster rate than either uninduced or FLAM3 KD cells, as determined by the change in OD600 after 6 hours (*p-v*alues = 0.009 and 0.004, two-tailed paired *t*-tests, respectively, Fig 4B). These results are consistent with previously reported cases of silencing several other ciliary proteins [30, 43, 48]. Together, the sedimentation assay results suggest that TbLC2 regulates ciliary motility to maintain effective swimming motility of trypanosome cells.

## Overexpression of TbLC2 only partially restores cell growth and sedimentation assay-determined motility defects

The tetracycline-inducible overexpression of recombinant TbLC2::eGFP using the pLEW100V5 plasmid system (Materials and methods) only partially restored the cell division (S7 Fig and S1 Text), growth (Fig 4A), and sedimentation (Fig 4B) phenotypes associated with TbLC2 knockdown in FLAM3 knockdown cells (i.e., FLAM3-LC2 KD/LC2 OE cells). We hypothesized that the partial restoration, rather than fully restoring to the uninduced or FLAM3 knockdown phenotypes, was due to overexpression of TbLC2::eGFP alone. Therefore, we further examined the phenotypes of WT/LC2 OE cells, which express the endogenous TbLC2 protein as well as the recombinant chimeric TbLC2::eGFP protein.

Using flow cytometry (Materials and methods), we found that the expression level of the recombinant LC2::eGFP was similar in the WT/LC2 OE and FLAM3-LC2 KD/LC2 OE cells, with both cell lines showing a higher fluorescence intensity than the background fluorescence signal shown by the WT cells (S3 Fig). We also found that TbLC2::eGFP overexpression in

FLAM3-LC2 KD/LC2 OE cells showed a small (1.6 fold) but significant (*p-value* = 0.012, two-tailed paired *t*-test) increase in the cell density as compared to FLAM3-LC2 KD cells after six days (Fig 4A). Additionally, the overexpression in FLAM3-LC2 KD/LC2 OE cells only partially restored the motility phenotype, as shown by the smaller change in the OD600 value as compared to FLAM3-LC2 KD cells (*p-value* = 0.023 at 6 hours, two-tailed paired *t*-test, Fig 4B) but a change in OD600 still significantly below the level of FLAM3 KD cells (*p-value* = 0.0041 at 6 hours, two-tailed paired *t*-test, Fig 4B).

Additionally, we found that the WT/LC2 OE cell line showed a significant growth defect, with a 2.6-fold decrease in cell density at the end of 6 days (*p-v*alue < 0.0001, two-tailed paired *t*-test, Fig 4A), as compared to wild-type cells, as well as significant cell division and kinetoplast localization phenotypes (S7 Fig and S1 Text). Moreover, WT/LC2 OE cells showed a significantly faster sedimentation rate (*p-v*alue < 0.0001 at 6 hours, two-tailed paired *t*-test, Fig 4B) than uninduced cells. Together, these results suggest that overexpression of TbLC2::eGFP, in addition to and in conjunction with knockdown of endogenous TbLC2, causes growth and motility defects.

## LC2 knockdown upregulates ciliary beat frequency

The length, beat frequency, and beating waveform are the primary biophysical properties of cilia that determine the swimming motility of ciliated microorganisms [18]. While having their cilium wrapped around their cell body causes wild-type *T. brucei* cells to exhibit out-of-plane, auger-like swimming that facilitates navigation through the host [3], the fusion of the cilium to the cell body, the out-of-plane nature of its beating waveform that results from that fusion, and the high frequency of ciliary beating make it impossible to extract meaningful data about the ciliary beating waveforms of wild-type trypanosome cells from the quantitative image analyses of our high-speed and high-resolution video microscopy. FLAM3 knockdown disrupts the trypanosome's FAZ, reducing the extent to which the cilium it is attached to the cell body [40], enabling us to quantify the above-mentioned biophysical properties of cilia (Materials and methods). Additionally, FLAM3 knockdown alters the mechanical constraints acting on the cilium, and thus the effective stiffness of the cilium's anterior, making the cells more susceptible to the effects of genetic perturbations, like TbLC2 knockdown, that affect ciliary beating biophysical mechanisms [47]. Therefore, we used the FLAM3 knockdowns in the characterization of ciliary length, beat frequency, beating waveform, and swimming motility in the paragraphs that follow.

We measured the lengths of FLAM3 KD and FLAM3-LC2 KD cilia and found that they were not significantly different (*p-value* = 0.32, two-tailed paired *t*-test, S7 Fig), suggesting that length cannot explain the effect of TbLC2 knockdown on the sedimentation assay-determined swimming motility phenotype (Fig 4B).

We also examined the beat frequency of the various TbLC2 knockdown strains. To do so, we used a single-beam optical tweezer because of its ability to get high resolution frequency data with cells that are minimally constrained (Fig 5A, Materials and methods). We found that the cells trapped at a single point displayed unconstrained ciliary beating behavior, with the cells performing a non-planar ciliary beat, like freely swimming cells (S1 and S2 Movies). We also found that the laser trapped both wild-type and FLAM3 knockdown cells, which have different body shapes and ciliary attachments (Fig 2), at a point approximately 2/3 of the cell length toward the posterior end (Fig 5B and S1 and S2 Movies). The specific location of the trapped position on the cells towards the posterior end suggests the cells have a localized structure suitable for optical trapping. Since the relative refractive index of the trapped material determines the intensity of gradient force exerted by the laser [49], DNA (mitochondrial,

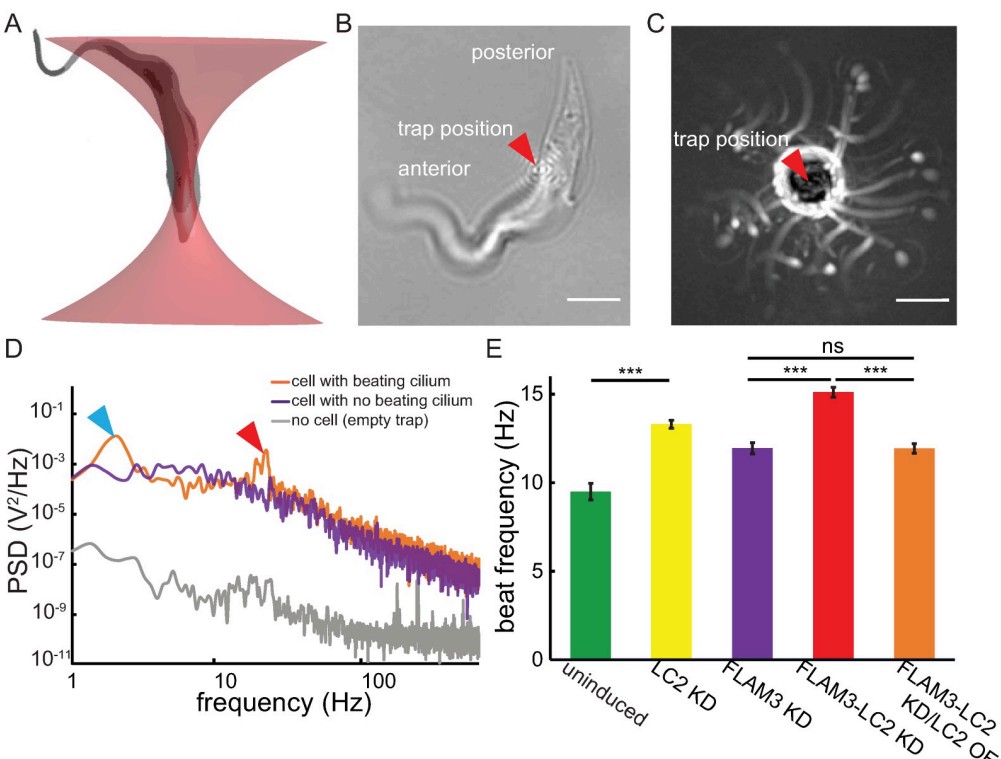

**Fig 5. Optically trapped LC2 KD and FLAM3-LC2 KD cells showed upregulated ciliary beat frequency. A**. Schematic of a *T. brucei* cell (*gray*) trapped by a low power (~20 mW) optical tweezer (*red*). **B**. Brightfield image of an uninduced FLAM3-LC2 KD cell taken near the coverslip, thus enabling direct visualization of the trap location and aligning the cell in the imaging plane. The identified trap location on a trypanosome cell (*red arrowhead*) is typical of all cells we trapped. Scale bar = 5 μm. **C**. Typical maximum projection (intensity inverted) of a movie (2 s) showing a trapped FLAM3 KD cell as it rotates about the trap center (*red arrowhead*). Scale bar = 5 μm. **D**. Typical power spectral densities (PSD) of a trapped cell with its cilium beating (FLAM3-LC2 KD/LC2 OE cell, *orange*), a trapped cell with its cilium not beating (FLAM3-LC2 KD/LC2 OE cell stalled at the bottom of the imaging chamber, *purple*), and background trap noise (*gray*, about six orders of magnitude smaller than the PSD of the cell with a beating cilium at the beat frequency). Peaks in the PSD represent the characteristic cell rotation rate (*blue arrowhead*) and ciliary beat frequency (*red arrowhead*). Each PSD represents the average of 3 spectra (1 second of data, each). **E**. Ciliary beat frequency, $f_\omega$, of multiple cell lines obtained from the higher of the two characteristic frequencies from the PSD analysis. The error bars represent the SEM and *** represents *p*-values < 0.0001 and ns represents p-values > 0.05 from two-tailed paired *t*-tests. N = 25, 86, 75, 85, and 96 for uninduced, LC2 KD, FLAM3 KD, FLAM3-LC2 KD, and FLAM3-LC2 KD/LC2 OE cells, respectively.

nuclear, or in the kinetoplast, in *T. brucei*'s case), which has a relatively high refractive index, in the cell body could determine the preferred trapping location [50].

We found that trypanosome cells trapped by the laser swim in a tight circle about the center of the trap (Fig 5C). We further analyzed their swimming using power spectral density (PSD) analysis (Materials and methods). We found that the typical PSD had two peak frequencies: $f_0$ at approximately 1–2 Hz and $f_\omega$ at approximately 10–15 Hz (Fig 5D). The lower frequency, $f_0$, represents the rotational motion of the cell swimming about the trap location, a motility mode that arises from the moment generated by the ciliary propulsive force acting about the trap center. We corroborated this interpretation of $f_0$ using maximum projections of time-series image sequences, which showed cells rotating about the trap location at this frequency (Fig 5C).

Due to the hydrodynamic forces acting on a free-swimming trypanosome cell with a cilium that wraps around the cell body, i.e., extensively in wild-type cells but only minimally in

FLAM3 knockdown trypanosome cells, forward swimming motility generates a rotation of a free swimming cell about its long body axis at a frequency of approximately 2 Hz (5–8 single ciliary beats per full rotation) [51]. However, with no net forward propulsion of cells under the optically trapped condition, the cells do not rotate about their long body axis [52, 53], which is commensurate with the lack of a third peak in the PSDs at about 2 Hz (Fig 5D). Therefore, we concluded that the higher frequency, $f_\omega$, corresponded to ciliary beat frequency.

We found that the beat frequency, $f_\omega$ from PSDs, of LC2 KD (13.3 ± 0.2 Hz, mean ± SEM) was significantly higher than the beat frequency of uninduced cells (9.5 ± 0.46 Hz, *p*-value < 0.0001 cases, two-tailed paired *t*-test, Fig 6E). Additionally, we found that $f_\omega$ from FLAM3-LC2 KD (15.1 ± 0.3 Hz) cells was significantly higher than FLAM3 KD cells (11.95 ± 0.31 Hz, *p*-value < 0.0001 in both cases, two-tailed paired *t*-test, Fig 5E). Overexpressing TbLC2::eGFP using the pLEW100V5 vector system in FLAM3-LC2 KD cells (FLAM3-LC2 KD/LC2 OE cells) restored the ciliary beat frequency to the parent FLAM3 KD cell line ciliary beat frequency (11.9 ± 0.3 Hz, *p*-value = 0.49, two-tailed paired *t*-test, Fig 5E). Together, these results show that knockdown of TbLC2 upregulates the beat frequency and thus suggests that TbLC2 downregulates the ciliary beat frequency in wild-type trypanosome cells, which is consistent with the regulatory role of other axonemal dynein associated proteins in *C. reinhardtii* [18] and *L. mexicana* [25].

## Ciliary motility is less directional in TbLC2 knockdown cells than wild-type cells

The combination of an elevated ciliary beat frequency (Fig 5E), which indicated knocking down TbLC2 upregulates motility, with an increased sedimentation rate (Fig 4B), which indicated knocking down TbLC2 downregulates motility, suggests a more complicated role for LC2 in regulating trypanosome cell swimming behavior, potentially involving the directional persistence of the swimming path. To investigate the role of TbLC2 in the directional persistence of trypanosome cell swimming paths, we recorded movies of freely swimming trypanosome cells (Materials and methods, S3, S4 and S5 Movies for examples). We analyzed the movies for individual cell swimming trajectories (Materials and methods) and plotted the 10-second trajectories from a single origin for various cell strains (Fig 6A). The longer swimming trajectories suggest that uninduced, LC2 KD, and FLAM3 KD cells exhibited more directionally persistent motility than FLAM3-LC2 KD, FLAM3-LC2 KD/LC2 OE, and WT/LC2 OE cells (Fig 6A). Most strikingly, we found that FLAM3-LC2 KD cells exhibited less directionally persistent phenotype than FLAM3 KD cells (Fig 6A).

A closer look into the video microscopy (S3, S4 and S5 Movies) showed that the shorter and less directionally persistent swimming trajectory phenotype associated with TbLC2 knockdown was characterized by cells that either mostly tumbled or swam short distances between tumbling events, as compared to the uninduced or FLAM3 KD cells. We also found that the LC2 KD and FLAM3-LC2 KD cells maintained a vigorous ciliary beat frequency (S3, S4 and S5 Movies and Fig 5E), despite the propensity to tumble. Together, these observations suggest a role for TbLC2 in regulating the directionality of ciliary-driven swimming motility in trypanosomes.

To quantify the differences in directional persistence of the trypanosome cells (Fig 6A), we calculated the mean of the magnitude of the average velocity, $|\vec{v}|$, which is the magnitude of the total displacement of the cell from its starting position to its final position, $|\Delta\vec{r}|$, divided by the time it took to undergo the displacement, $\Delta t$ ($|\vec{v}| = \frac{|\Delta\vec{r}|}{\Delta t}$, Materials and methods). We found that LC2 KD cells (4.24 ± 0.20 μm/s, mean ± SEM) swam with slightly (1.1-fold), but not significantly (*p*-values = 0.14, two-tailed paired *t*-test, Fig 6B), reduced average velocity than

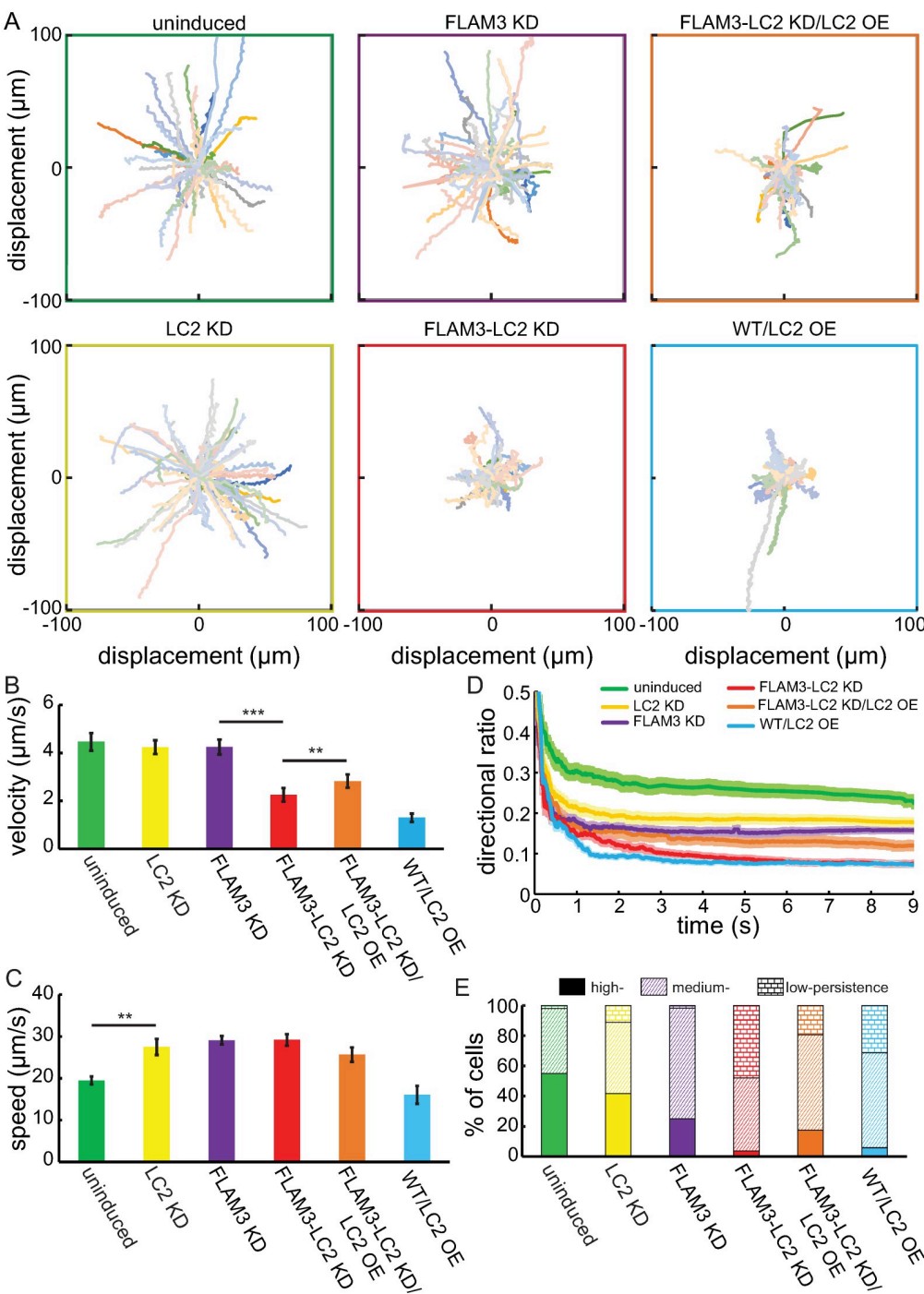

**Fig 6. LC2 knockdown trypanosome cells have non-processive swimming behaviors. A**. Swimming trajectories of trypanosome cells plotted from an initial position at the origin. Each trajectory represents 10 seconds of tracked data. **B**. Magnitude of the average velocity ($|\vec{v}| = \frac{|\Delta \vec{r}|}{\Delta t}$, Materials and methods) of swimming trypanosome cells. **C**. Mean average swimming speed ($s = \frac{L}{\Delta t}$, Materials and methods) measured for multiple cell lines over the entire trajectory of 10 s. Error bars represent the SE of the mean, $^{***}$ corresponds to $p$-values $< 0.0001$, and $^{**}$ corresponds to $p$-values $< 0.003$, two-tailed paired $t$-tests, in panels A. and B. **D**. The mean directional ratio (DR, dark lines) calculated over elapsed time and averaged for each strain. Error bars (lightly shaded bands) represent the SE of the mean. **E**. The percentage of cells that exhibit high persistence (DR $> 0.2$, solid color), medium persistence ($0.05 < $ DR $< 0.2$, hatch mark), and low persistence (DR $< 0.05$, brick pattern), as determined by the last point (as time goes to infinity) directional ratio. We showed paths for and calculated values from N = 50, 75, 53, 73, 47, and 68 for uninduced, LC2 KD, FLAM3 KD, FLAM3-LC2 KD, FLAM3-LC2 KD/LC2 OE, and WT/LC2 OE cells, respectively, in all panels.

uninduced cells (4.62 ± 0.37 μm/s, mean ± SEM), in accordance with our observations of trajectory length (Fig 6A). We also found that the LC2 knockdown reduced average velocity phenotype was amplified in FLAM3 knockdown cells, with FLAM3-LC2 KD cells (1.79 ± 0.18 μm/s, mean ± SEM) exhibiting a 2.5-fold reduced average velocity as compared to FLAM3 KD cells (4.48 ± 0.30 μm/s, mean ± SEM, *p*-value < 0.0001, two-tailed paired *t*-test, Fig 6B). Additionally, we found that uninduced and FLAM3 KD cells had similar average velocities (*p*-value = 0.77, two-tailed paired *t*-test, Fig 6B), as previously reported [54].

We also calculated the mean of the average swimming speed, *s*, which is the integrated path length of the cell along its track, *L*, divided by the time it took to traverse that distance ($s = \frac{L}{\Delta t}$, Materials and methods), of multiple cell lines. We found that the average speed was higher for LC2 KD (27.5 ± 1.93 μm/s) than uninduced cells (19.5 ± 0.9 μm/s, *p*-value = 0.0014, which was in good agreement with the reported average speed of wild-type trypanosome cells [24, 54–56], two-tailed paired *t*-test, Fig 6C). The FLAM3 KD and FLAM3-LC2 KD cells (29.1 ± 1.0 μm/s, 29.2 ± 1.4 μm/s, respectively) exhibited higher average speed than uninduced cells (*p*-value < 0.0001 in both cases, two-tailed paired *t*-tests, Fig 6C) but not significantly different from each other (*p*-value = 0.95).

## LC2 knockdown causes trypanosome cells to swim with lower directional persistence

The faster average swimming speed of LC2 KD as compared to uninduced cells (Fig 6C) likely precludes the possibility of significant defects in ciliary beat generation explaining the reduced cell motility (Fig 6A). Additionally, the significantly lower average velocity than average swimming speed in the LC2 knockdowns suggests a highly futile swimming behavior with low directional persistence. We used the directional ratio, DR, defined as the ratio of displacement to path length (Materials and methods [57]), to quantify trypanosome cell motility directional persistence. In accordance with the degree of the discrepancy between the average swimming speed and average velocity measurements, the LC2 KD cells exhibited a lower directional ratio at the final point of the swimming trajectory (DR = 0.18 ± 0.01, mean ± SE of the mean, Fig 6D) as compared to uninduced cells (0.23 ± 0.01, *p*-value = 0.004, two-tailed paired *t*-test, Fig 6D). Moreover, FLAM3-LC2 KD cells exhibited a lower directional ratio at the final point of the swimming trajectory (DR = 0.07 ± 0.01, mean ± SE of the mean, Fig 6D) as compared to FLAM3 KD cells (0.16 ± 0.009, *p*-value < 0.0001, respectively, two-tailed paired *t*-test, Fig 6D).

A highly-regulated forward (tip-to-base) ciliary beating mode maintains the persistent swimming motility of trypanosome cells. Disruption to such regulation leading to abrupt switching of the ciliary beating mode leads to cellular reorientation (tumbling) [24]. To characterize the relative occurrence of tumbling in the population, we binned swimming cells into three directional persistence classes, high persistence (DR > 0.2, persistent swimmers, S3 Movie), medium persistence (0.05 < DR < 0.2, persistent swimmers with intermittent tumbling, S4 Movie), and low persistence (DR < 0.05, tumblers, S5 Movie), based on their directional ratio at the end of the 10-second tracks (Fig 6D).

We found that approximately 50% of the uninduced cell population exhibited high persistence while very few exhibited low persistence (Fig 6E), which is consistent with previous reports [24]. LC2 KD cells showed a significant increase in the low persistence swimmer population to 11% (*p*-value = 0.043, two-tailed paired *t*-test, Fig 6E) and a corresponding decrease in the high persistence swimmer population to 42% (*p*-value = 0.11, two-tailed paired *t*-test, Fig 6E), as compared to uninduced cells. The knockdown of FLAM3, alone, also caused a significant decreased in the fraction of high persistence swimmers to approximately 25% (*p*-value = 0.0018, two-tailed paired *t*-test, Fig 6E) without significantly changing the fraction of

low persistence swimmers (*p*-value = 0.97, two-tailed paired *t*-test, Fig 6E), as compared to uninduced cells. However, the FLAM3 knockdown phenotype amplified the effects of TbLC2 knockdown in FLAM3-LC2 KD cells, which showed a significant increase in the low persistence swimmer population to 48% (*p*-value < 0.0001, two-tailed paired *t*-test, Fig 6E) and a corresponding decrease in the high persistence swimmer population to 3.8% (*p*-value = 0.0005, two-tailed paired *t*-test, Fig 6E) as compared to un FLAM3 KD cells.

Together, the motile cell trajectory results (Fig 6) suggest that TbLC2 regulates axonemal dynein in a way that reduces the tumble bias of trypanosome cells. They suggest that TbLC2 stabilizes the switching mechanism between forward propulsion, with tip-to-base bending wave propagation along the cilium, and tumbling-like cell reorientation, with base-to-tip bending wave propagation. This behavior is reminiscent of mechanisms that alter the tumble bias in run-and-tumble swimming behavior [2, 58].

## Overexpression of TbLC2 leads to slow and futile ciliary motility

With the TbLC2 overexpression in cells already shown to be linked to growth and cell division phenotypes (Figs 4 and S7), we further examined the effect of the overexpression in ciliary motility. We found that overexpression of TbLC2::eGFP in WT/LC2 OE cells showed a 3.5-fold reduction (*p*-value < 0.001, two-tailed paired *t*-test, Fig 6B) in the average velocity as compared to uninduced cells. Additionally, we found that the overexpression of TbLC2::eGFP in FLAM3-LC2 KD/LC2 OE cells showed partial, rather than total, restoration of average velocity shown by 1.4-fold increase in average velocity of the FLAM3-LC2 KD cells (*p*-value = 0.003, two-tailed paired *t*-test, Fig 6B) that amounts to only 57% of that of the FLAM3 KD cells (*p*-value < 0.0001, two-tailed paired *t*-test, Fig 6B).

Moreover, a significantly larger fraction of FLAM3-LC2 KD/LC2 OE cells were highly persistent swimmers (17.4%, *p*-value = 0.011, two-tailed paired *t*-test, Fig 6E), and a lower fraction were low persistence swimmers (19.6%, *p*-value = 0.0016, two-tailed paired *t*-test, Fig 6E), as compared to the FLAM3-LC2 KD cells. Complementing the growth and sedimentation assay motility phenotype results (Fig 4), a majority of the WT/LC2 OE cells showed medium or low persistence swimming behavior with a significant decrease in the high persistent swimmer proportion compared to the uninduced cells (5.9% from 55.1%, *p*-value < 0.001, two-tailed paired *t*-test, Fig 6E). Together, these results suggest that there is an optimal LC2 expression level in cells and that effects due to overexpression of recombinant TbLC2, alone, might explain the lack of motility phenotype restoration in FLAM3-LC2 KD/LC2 OE cells, as compared the FLAM3 KD cells.

## LC2 and FLAM3 double knockdown alters the ciliary waveform of trypanosome cells

The combination of increased average swimming speed (Fig 6C) and beat frequency (Fig 5E) with lower total displacement (Fig 6A), average velocity (Fig 6B), directionality ratio (Fig 6D), and no change in the overall length of the cilia (S6 Fig) all suggest that the effects of TbLC2 on ciliary motility may be most significantly tied to TbLC2's role in regulating the shape of the ciliary waveform and its role in the directionality of ciliary motility. The results suggest this, in part, due to the increased likelihood of TbLC2 knockdown cells exhibiting the low-persistence tumbling motility mode (Fig 6E), which is often associated with switching between tip-to-base to base-to-tip ciliary bending wave propagation swimming modes (S6 and S7 Movies) [3, 47].

Therefore, we tracked individual beating cilia and quantified the shape of the ciliary beating waveform from multiple FLAM3 KD and FLAM3-LC2 KD cells (Materials and methods) to probe the underlying biophysical mechanisms effect on the characteristic non-directional

swimming behaviors of trypanosome cells. We first calculated the tangent angle ($\psi$, S9 Fig, *inset*) of the cilium as a function of normalized arc length along the cilium's length, (s/L, S9 Fig), and then calculated the normalized curvature ($\kappa$/L) by taking the derivative of the tangent angle with respect to normalized arc length (Materials and methods) and plotted it as a function of normalized arc length (Fig 7A and 7B). We found that the beat waveforms of both FLAM3 KD and FLAM3-LC2 KD cells were asymmetric, i.e., the maximum positive curvature ($\kappa_{+,max} = \frac{1}{r_1}$, Fig 7C) was greater than the maximum negative curvature ($\kappa_{-,max} = \frac{1}{r_2}$, Fig 7C) for a beat period. We calculated the maximum curvature asymmetry ratio, $AR_{MC}$, as the ratio of $\kappa_{+,max}$ to $\kappa_{-,max}$ (Materials and methods), and found that $AR_{MC}$ was larger in the FLAM3-LC2 KD cells (1.35 ± 0.03) than FLAM3 KD cells (1.11 ± 0.06, *p*-value = 0.0006, two-tailed paired *t*-test, Fig 7D). Additionally, we found that the FLAM3-LC2 KD cells' ciliary beat exhibited reduced peak-to-peak tangent angle amplitude (1.11 ± 0.03 radians, mean ± SEM, S9 Fig) as compared to FLAM3 KD cells (1.25 ± 0.02, *p*-value = 0.004, two-tailed paired *t*-test, N = 25 beating periods from approximately ten cells, S9 Fig). However, the beat wavelength as quantified by the normalized arc length between two consecutive tangent angle peaks along a beating cilium (S9 Fig) remained unaltered (0.96 ± 0.03 and 0.99 ± 0.05, *p*-value = 0.62, two-tailed paired *t*-test, N = 25 beating periods from approximately ten cells) between the FLAM3 KD and FLAM3-LC2 KD cell lines, respectively.

Cilia that exhibit a non-zero mean curvature (S9 Fig) in combination with a symmetric dynamic curvature (S9 Fig) propel straight-line swimming when the beats are synchronized in biflagellate cells, like *C. reinhardtii* [23], and lead to circular or spiral swimming trajectories in uniflagellate cells, like mammalian sperm [23]. Since *T. brucei* have a single cilium, effects that increase the asymmetry of either the static or forward swimming tip-to-base dynamic curvature could lead to curved swimming paths that reorient the cell body as it swims [23]. We found that the intermittent base-to-tip ciliary beat mode of trypanosomes was highly asymmetric, with large maximum curvature bending wave propagation (S8 Movie and S10 Fig). Furthermore, FLAM3-LC2 KD cells showed a more highly asymmetric, non-zero static component of the ciliary beating waveform (S11 Fig) during a base-to-tip (reverse) ciliary beating than during the tip-to-base (forward) beating. Both of these observations correlated with the reorientation of the cell's swimming direction upon switching of its forward swimming tip-to-base beating to its reverse bast-to-tip mode (S6 and S7 Movies, Fig 6A and 6D).

To examine whether the LC2 knockdown shifted the bias of the ciliary beating mode, we investigated the fraction of time the cells dwelled in the reversed base-to-tip beating mode versus the forward tip-to-base mode (Materials and methods). We found that FLAM3-LC2 KD cells dwelled in the base-to-tip beating mode approximately twice as long as FLAM3 KD knockdown cells (0.24 ± 0.027 and 0.10 ± 0.02, respectively, *p*-value < 0.001, two-tailed paired *t*-test, Fig 7E). We further analyzed these fractions by quantifying the duration (dwell time) of the individual ciliary beat mode events (Fig 7F) and fitting exponential functions to the cumulative probability distribution functions of the dwell times (Table 1).

We found that the FLAM3-LC2 KD cells dwelled approximately 2-fold longer in each reverse base-to-tip event and 1.7-fold shorter in the forward tip-to-base beating mode events than FLAM3 KD cells (*p*-value = 0.0025 and *p*-value < 0.0001, two-tailed paired *t*-tests, respectively). Furthermore, although the TbLC2 overexpression restored the symmetry of the ciliary beating waveform to that of FLAM3 KD (*p*-value = 0.42, two-tailed paired *t*-tests, Fig 7D), the overexpression had minimal effect on the forward and reverse beating dwell times compared to that of the FLAM3-LC2 KD cells (*p*-value = 0.023 and *p*-value = 0.12, two-tailed paired *t*-

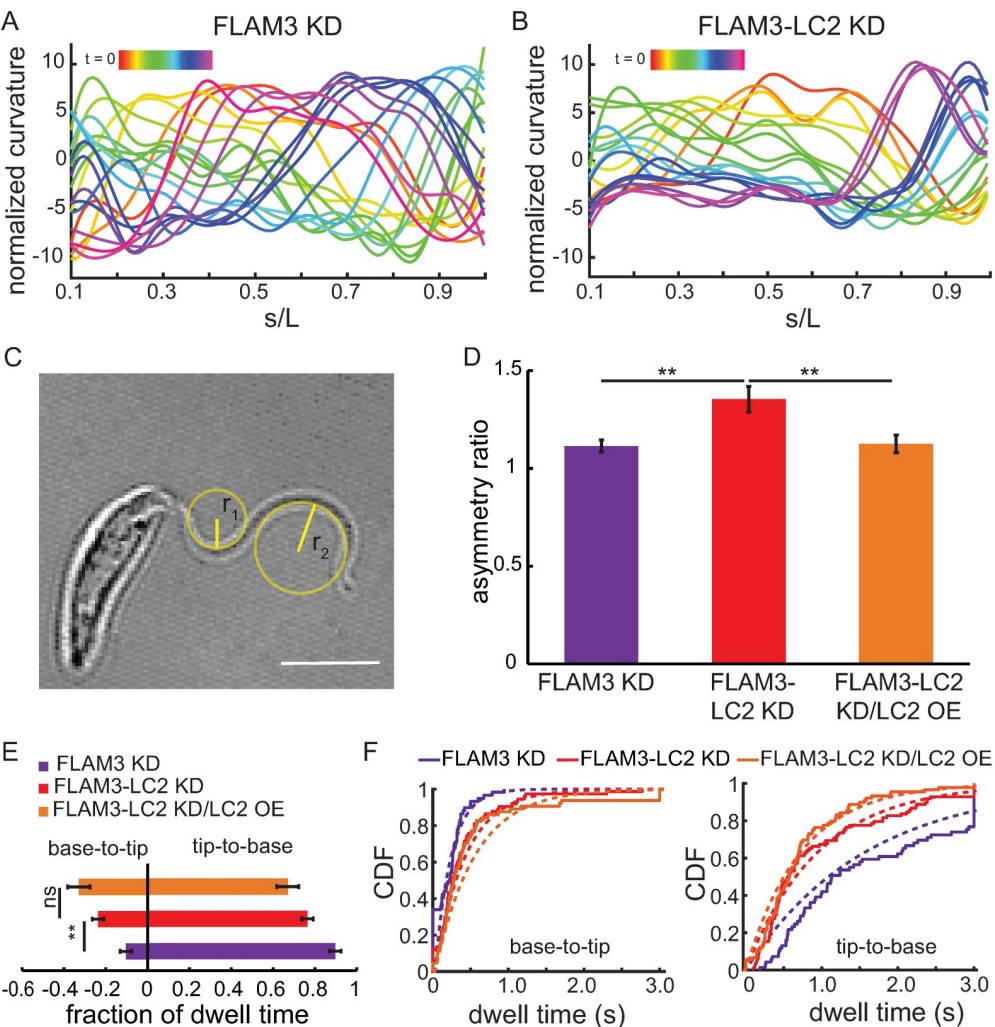

**Fig 7. Loss of TbLC2 alters multiple aspects of the ciliary beat waveform.** Representative plots of curvature, κ, normalized to the contour length (L) for **A**. FLAM3 KD and **B**. FLAM3-LC2 KD cells plotted as a function of s/L, which is the ratio of the arc length along the cilium (s) to the total ciliary length (L), where s/L = 0 is the base and s/L = 1 is the tip of the cilium. Colors (*red* to *magenta*) represent the time evolution of ciliary beat shape over one complete period of the oscillation. **C**. Schematic showing the radii of curvature corresponding to the maximum positive, $r_1$, and the maximum negative, $r_2$, curvature, where the curvature is 1/the radius of curvature, κ = 1/r, in this frame. We used these radii of curvature to calculate the asymmetry ratio of a ciliary bend during a ciliary beating. This image is a representative frame from a high-speed movie of a FLAM3-LC2 KD cell. Scale bar = 5 μm. **D**. Mean maximum curvature asymmetry ratio, $AR_{MC}$, for FLAM3 KD, FLAM3-LC2 KD, and FLAM3-LC2 KD/LC2 OE cells. N = 15 waveforms from 10–12 different cells in each cell line, the error bars represent the SEM, and ** represents a *p*-value < 0.005, two-tailed paired *t*-tests. **E**. Fraction of time cells dwelled in tip-to-base (positive fraction) and base-to-tip (negative fraction) ciliary beating modes. N = 42, 44, and 33 cells for FLAM3 KD, FLAM3-LC2 KD, and FLAM3-LC2 KD/LC2 OE cells, respectively. ** represents a *p*-value of < 0.001 and ns a *p*-value > 0.05, two-tailed paired *t*-tests. **F**. Cumulative distribution function (CDF) of individual dwell times for various FLAM3 knockdown cells in base-to-tip (*left*) and tip-to-base (*right*) ciliary beating modes. The fits (*dotted lines*) to exponential functions are shown. Forty-five cells of each strain were analyzed with a total number of n = 59, 75, 65 base-to-tip mode and n = 70, 99, and 90 tip-to-base mode switching events for FLAM3 KD, FLAM3-LC2 KD, and FLAM3-LC2 KD/LC2 OE cells, respectively.

**Table 1. Characteristic beating mode dwell times.** Dwell time is the fit parameter ± the standard error of the fit, obtained by fitting exponential functions to the cumulative probability distributions of the dwell times (Fig 7F).

| Cell lines | Dwell time (forward beat) (s) | Dwell time (reversed beat) (s) |
|---|---|---|
| FLAM3 KD | 1.57 ± 0.19 | 0.21 ± 0.03 |
| FLAM3-LC2 KD | 0.95 ± 0.10 | 0.40 ± 0.04 |
| FLAM3-LC2 KD/LC2 OE | 0.73 ± 0.08 | 0.52 ± 0.07 |

tests, respectively for forward and reverse beat dwell time, Fig 7F) reiterating the requirement of optimal LC2 expression level in the cell.

These results showed that the TbLC2 knockdown cells more rapidly switched into the base-to-tip (reverse) beating mode and were slower to switch back to the tip-to-base (forward) beating mode than those cells without TbCL2 knocked down. This result suggests that TbLC2 promotes directional persistence in trypanosome swimming paths by biasing cilium beating mode switching mechanism toward the forward swimming, tip-to-base beating mode.

## Discussion

In this study, we identified a *Trypanosoma brucei* homolog of *Chlamydomonas reinhardtii* LC2 (Fig 1), a Tctex-type axonemal dynein light chain that we named TbLC2. We demonstrated that TbLC2 localizes to the axoneme (Fig 2) and regulates various cellular and ciliary functions without being required for dynein assembly into the axoneme (Fig 3). We found that TbLC2 knockdown impaired cellular swimming motilities (Fig 4B) and lowered directional swimming persistence (Fig 6). These phenotypes correlated with higher cilium beating frequencies (Fig 5), increased static (time-averaged) curvature (Fig 7), greater dynamic curvature asymmetries (Fig 7), and a shift in beat mode bias away from the forward motility tip-to-base bending wave propagation mode (Fig 7). Thus, we propose that TbLC2 keeps the ciliary beat of trypanosomes under control. Without the regulatory effects of TbLC2 on axonemal dyneins within the cilium, trypanosome cells are going nowhere fast.

### TbLC2 directly regulates axonemal dynein

When all taken together, the observations and results we have made in this study suggest that TbLC2 tunes the waveform of beating trypanosome cilia and thus is necessary for highly persistent, directional cell swimming motility by directly regulating axonemal dynein. To physically change the waveform or frequency of a beating cilium a regulation mechanism must either alter the behavior of the force generators (and the only active force generating elements that bend axonemes are axonemal dyneins), alter the effective stiffness of the axoneme (or other structures like the paraflagellar rod), alter the effective internal viscosity of the axoneme, or alter the effective stokes drag coefficient of the cilium as it moves through the surrounding fluid [59]. It is highly unlikely that TbLC2 regulates structural (stiffness) elements, the internal viscosity, or the stokes drag of the axoneme. Therefore, our results suggest that TbLC2 regulates the force-generating axonemal dynein motor proteins.

Our suggestion that TbLC2 affects motility by regulating dynein is generally consistent with studies on other axonemal dynein-associated proteins. For example, this interpretation is consistent with results showing that the knockdown of [30] and mutations to [60] outer arm dynein-associated LC1 and outer arm dynein-associated intermediate chain DNAI1 [61] shifted the waveform propagation mode bias toward reverse swimming motility base-to-tip ciliary beating and led to the formation of cell clusters due to abnormal cytokinesis in *T. brucei*. Similarly, axonemal dynein-associated protein LC1 also regulates the mechanochemical cycles

of outer arm dynein and maintains directional swimming persistence in *Chlamydomonas* cells [20]. Additionally, the asymmetry of proximal/distal outer arm dynein molecular composition regulates the beating waveform propagation direction in *T. brucei* and *L. mexicana* [25]. Deleting the LC4-like outer arm dynein light chain increased the ciliary beat frequency, average swimming speed, and average velocity of *L. mexicana* [25]. Similarly, disruption of Tcte3-3 outer arm dynein light chain increased the beat frequency mouse spermatozoa [27]. Moreover, our bioinformatic (S1 and S2 Figs) and structural (Fig 1) analyses show significant homology between TbLC2 and *Chlamydomonas*'s outer arm dynein associated LC2. The aggregate of these results and their correspondence to our results further suggest that TbLC2's mechanism likely acts on outer arm dynein, specifically.

Assuming trypanosome dynein complexes share structural homology with *Chlamydomonas* [26] and *Tetrahymena thermophila* [62, 63] axonemes, our bioinformatic and structural data (Figs 1 and S1) suggest that TbLC2 likely localizes to the outer arm dynein light chain tower, a considerable distance (15 nm) from outer arm dynein's ATP hydrolysis site. Given this distance, our results suggest that TbLC2 regulates dynein function through an intricate network of LC-LC and LC-IC interactions or perhaps with the next nearest neighboring dynein motor along the axoneme, based on its putative location in the dynein complex [26]. Furthermore, this analysis suggests that TbLC2 may directly interact with the HC-IC complex as a heterodimer with the trypanosome homolog of *Chlamydomonas* LC9 [26]. Additionally, a recent crosslinking experiment showed that *Chlamydomonas* LC9, in turn, associates with IC1 and IC2 and thus forming an LC-IC block [64] that influences the function of outer arm dynein in *Chlamydomonas* cilia through an interaction with the dynein heavy chains. In sum, these results from *C. reinhardtii* suggest that TbLC2, also being a Tctex-like light chain, might be an indispensable piece of IC-LC block architecture responsible for regulating the activity of outer arm axonemal dynein.

However, we cannot rule out a role for TbLC2 with inner arm dynein, as well. In *Chlamydomonas* cells, the loss of inner arm dynein g-interacting dynein regulatory complex protein DRC3, a component of the nexin dynein regulatory complex (N-DRC) [18], elevated the average ciliary beat frequency. Moreover, we found both frequency and waveform phenotypes, which are classically associated with outer arm dyneins and inner arm dyneins, respectively [65, 66]. The correspondence of inner arm dynein regulator phenotypes to our results combined with the lack of evidence ruling out the possibility of an inner arm dynein localization of TbLC2 (Figs 1, 2 and 3) does not enable us to rule out that TbLC2 also associates with and regulates inner arm dynein in trypanosome cells.

Some knockouts, knockdowns, or mutations of axonemal dynein-related light chains, including LC1 in *Chlamydomonas* [20] and trypanosomes [30], and LC2 in *Chlamydomonas* [43], have shown some contrasting motility phenotypes to the ones reported in this study, including a reduced ciliary beat frequency. However, the deletions in these cases led to failures in outer arm dynein assembly. We found no noticeable defects in the axonemal structures, including the outer and inner arm dyneins, with LC2 knockdown (Fig 3). Therefore, further direct comparison to these other cell biophysical phenotypic studies of axonemal dynein-associated light chains is difficult. Our results suggest that TbLC2 is not required for the stable assembly and activation of dyneins in the axoneme, the preassembly of axonemal dynein complexes in the cytoplasm, or the transport of dynein complexes to the axoneme.

## Overexpression of recombinant LC2 is sufficient to cause growth and motility defects

Overexpression of the recombinant TbLC2 in both the wild-type (WT/LC2 OE) and FLAM3-LC2 KD (FLAM3-LC2 KD/LC2 OE) cells resulted in its localization to the cilium

(Figs 2D, 2E and S5), in addition to the cytoplasm (Fig 2A). However, overexpression of the recombinant TbLC2 caused growth and motility defects (Figs 4 and 6), including that most WT/LC2 OE cells showed extended ciliary beat reversals (S9 Movie) and only partially restored the cell division, growth, and motility phenotypes in FLAM3-LC2 KD/LC2 OE cells (Figs S7, 4A, 4B, 6 and 7E) despite proper assembly of dynein into the axoneme (Fig 3) and efficient incorporation of TbLC2::eGFP into the cilium (Figs 2D, 2E and S5). When taken together, these WT/LC2 OE results suggest that there may be an optimal expression level of TbLC2 in trypanosome cells.

Several factors could contribute to these results, including the effect of the overexpression itself on the overall cell physiology and the effect of the C-terminal chimeric tags (eGFP, His6, and BCCP) on the secondary structure, electrostatic surface potential distribution, and potential introduction of steric interactions with other axonemal proteins. Overexpression of various other cilium-associated proteins, including the intraflagellar transport kinesin, TbKin2b, and inner arm dynein, IAD-1α, caused abnormalities in the formation of the cilium, the adherence of the cilium to the cell body, and the overall morphology of the cell, all possibly leading to slow growth rate [67]. Moreover, an HA tag on the dynein microtubule-binding domain-associated trypanosome outer arm dynein light chain 1 (LC1) hindered the function of dynein [10]. However, we do not expect such a severe effect of tags on the function of TbLC2 because it is unlikely to act directly at the interface between dynein and the microtubule, unlike LC1 [26], but further analysis on the structural and functional implications of tagging TbLC2 would be needed to rule out that possibility. In any of the possibilities mentioned above, the overexpression of recombinant TbLC2 in FLAM3-LC2 KD double knockdown cells was still sufficient to at least partially restore nearly all the phenotypes discussed above.

## Ciliary beat asymmetry, rather than waveform propagation direction, underlies changes in trypanosome cell swimming direction

We found that WT/LC2 OE cells spent long periods of time in the reverse base-to-tip ciliary beating mode (S9 Movie). However, they tended to swim with relatively high directional persistence, albeit backward (S9 Fig and S8 Movie) and seem to exhibit a low static curvature, symmetric base-to-tip ciliary beating mode (S10 Fig, S8 and S9 Movies). Unfortunately, we could not quantify these comparisons because performing the quantitative ciliary waveform tracking requires cell lines with detached cilia. Together, these observations suggest that the *asymmetry* of the beat, rather than the base-to-tip bending wave propagation direction, underlies the tumbling motility mode.

Furthermore, the higher ciliary beat frequency exhibited by TbLC2 knockdown cells (Fig 5E) should increase their swimming speed and directional persistence, in the absence of other effects. However, the higher asymmetry ratio exhibited by TbLC2 knockdown cells (Fig 7D) should decrease the swimming speed and persistence of cells, in the absence of other effects. Because the net effect on TbLC2 knockdown cells is a decrease in their directional motility persistence (Fig 6D and 6E), these results further support the conclusion that asymmetry of the beating waveform is the dominant effect of TbLC2 knockdown on swimming motility. Moreover, that the reverse base-to-tip ciliary beating mode in wild-type and TbLC2 knockdown trypanosomes is more asymmetric than the forward tip-to-base beating mode [22, 68] (Fig 7A and 7B as compared to S9 and S10 Figs) provides additional support for the conclusion that the asymmetry of the base-to-tip ciliary beat waveform dominates the tumbling mechanism, rather than the base-to-tip waveform propagation direction, itself.

The conclusion that the asymmetry of the base-to-tip ciliary beat waveform dominates the tumbling mechanism is also consistent with the observation that the highly asymmetric,

breaststroke-like ciliary beating of *Chlamydomonas* cells causes the cell body to exhibit minimal swimming persistence, i.e. to rotate, in absence of the balancing torque provided by the second cilium when the first has been ablated [69] or is missing due to a mutation to the basal body protein, Uni1 [70]. Moreover, swimming persistence is restored to *Uni1* mutant *Chlamydomonas* cells when subjected to a light shock [71] and the ciliary beat mode switches to a highly symmetric, sperm-like waveform [70]. Other *Trypanosomatidea* like *Leishmania major* and *Angomonas deanei* also show intermittent, highly asymmetric, breaststroke-like base-to-tip ciliary beating that causes cell reorientation and a significantly less efficient translational motility [22] than symmetric beating modes. Thus, we suggest a model of trypanosome directional motility that relies on a reversal of beating waveform propagation direction to switch the ciliary beat from a symmetric to an *asymmetric* waveform, which is the principal biophysical effect leading to the reorientation of the cell.

## *T. brucei* offers an attractive model system for studying mechanisms of ciliary motility in uniciliates

*T. brucei* has emerged as an excellent model organism for studying the cell biophysical aspects of eukaryotic cilia and flagella and molecular biophysics of axonemal dynein-based ciliary motility. Ciliary motility is an intricate part of the parasite's life cycle, so it provides a platform to understand how a myriad of molecules, including axonemal dyneins and dynein-associated protein regulators, underlie processes like cell division, swimming motility, and ultimately virulence. Trypanosome's tip-to-base beating mechanism, as well as its genetically accessible nature, make it a particularly compelling model to answer fundamental questions about ciliary beating. Moreover, its native attachment to the cell body and the ability to detach it with FAZ protein knockdown enable one to test fundamental biophysical theories about ciliary motility under the mechanical constraints imposed by the attachment to the cell body or without those effects, which makes the cilium more susceptible to environmental and genetic perturbations, as discussed above. FAZ protein knockdown also enables more straightforward biophysical characterizations of ciliary motility properties because imaging the shape of the beating waveform is easier when it is not bound to the cell body. In this study, we used each of these advantages to provide a quantitative analysis of how TbLC2 regulates multiple facets of trypanosome cell motility.

## Conclusion

In summary, we identified TbLC2 as a ciliary dynein light chain of the Tctex1/Tctex2 family that is necessary for various aspects of ciliary motility in trypanosome cells. Our work demonstrates that TbLC2 regulation of axonemal dynein causes the modulation of the ciliary beat and ultimately leads to cell-scale trypanosome-specific motility behaviors. Specifically, we found that TbLC2 knockdown cells had reduced directional swimming persistence characterized by a higher beat frequency, ciliary bending waveform defects, and a shift in the waveform propagation direction bias from the tip-to-base to the base-to-tip beating mode. Because knocking down TbLC2 did not cause significant structural phenotypes in the trypanosome cilia but did cause the functional phenotypes similar to, however distinct from, other species, our data, with the specific example of FLAM3 knockdown cell background, suggest that TbLC2 regulates axonemal dynein and the tip-to-base beating waveforms characteristic of *Trypanosoma brucei* ciliary motility. Moreover, our results suggest that TbLC2 downregulates and coordinates the underlying activity of axonemal dyneins in the cilium. These molecular mechanisms lead to the stabilization and symmetry of the oscillatory beating waveforms, and the

modulation of switching between the reverse and forward ciliary beat modes. Ultimately, TbLC2 helps to maintain the persistent directional motility of trypanosome cells.

The work provides the first evidence that TbLC2 is a key regulator dynein-driven ciliary motility in the disease-causing trypanosome. Further in vivo cell-scale research could elucidate TbLC2's role in providing trypanosome cells the ability to navigate various microenvironments in the tsetse fly vector [72], avoid the immune system in the mammalian host [9], and cross the blood-brain barrier [3], ultimately causing the most critical stage of disease. Additional in vitro single-molecule research could provide details on the fundamental molecular and biophysical mechanisms of TbLC2-related dynein regulation and flagellar waveform generation. We anticipate that a better understanding of the unique ciliary motility mechanisms in trypanosomes could eventually lead to the development of pan Kinetoplastea therapeutics that target unique aspects of their motility mechanisms.

## Materials and methods

### Sequence-based structural modeling of TbLC2

We identified putative *Trypanosoma brucei* homologs of *Chlamydomonas reinhardtii* LC2 (CrLC2, Cre12.g527750 [29]) using protein BLAST (http://www.ncbi.nlm.nih.gov/BLAST/ [35]). We considered proteins with $> 70\%$ coverage, expected values (E value) $< 1 \times 10^{-5}$, and a total alignment score $> 50$ to be good matches. We confirmed the ciliary localization of identified the proteins in the TrypTag database (tryptag.org) [36, 37]. We performed a multiple sequence alignment of the putative TbLC2 with the *C. reinhardtii* LC2 and its homologs in various other species (S1 Fig) using the multiple sequence alignment tool in Clustal Omega [73]. We performed the color-coding of the sequences based on the level of sequence conservation on Jalview 2.11.1.4 [74] by keeping the conservation index threshold at six [75].

We generated a sequence-based structure homology model of TbLC2 using CrLC2 (PDB ID: 7kzn [26]) as a template in the SWISS-MODEL (https://swissmodel.expasy.org) [38, 76, 77]. The cryo-EM CrLC2 structure lacked 22 N-terminal amino acids, likely due to the flexibility of this domain [26]. Therefore, the results from SWISS-MODEL had a similar N-terminal truncation (Fig 1). We also generated a sequence-based, template-free, AI-derived structural model using AlphaFold [39]. To directly compare with the SWISS-MODEL and cryo-EM structures, we used the same 22 amino acid N-terminal truncation in the sequence for AlphaFold.

We performed the 3D visualization of the modeled structures, the structural alignment, and the root mean square deviation (RMSD) calculations using PyMol (PyMol 2.3.4, Schrodinger, LLC). For optimal alignment of the structures, we used an RMSD outlier rejection cutoff of 2 Å and reported the final RMSD conferring the optimal alignment in the results. We used the Adaptive Poisson-Boltzmann Solver (APBS) [78] in PyMol to calculate the electrostatic potential and map it to the molecular surface. We compared the 22 amnio acid N-terminal truncated AlphaFold structure with the full-length AlphaFold predicted structure (S12 Fig), and we found the differences to be small (root mean square deviation, RMSD = 0.25 Å, S12 Fig).

We determined the residues at the CrLC2-LC9 binding interface by calculating the difference in accessible surface area between the CrLC2/LC9 complex and isolated CrLC2 and CrLC9 proteins (dASA) using PyMol. We set the cutoff for identifying interface residues at 1 $\text{Å}^2$ (a cutoff of 0 would mean all the residues are included as interface residues).

### Cloning GFP-tagged TbLC2 constructs

We adopted a well-developed strategy to introduce epitope tag-encoding DNA into specific target loci in the trypanosome genome using homologous recombination [67, 79]. We

extracted the *T. brucei* 29–13 genomic DNA (E.Z.N.A Tissue DNA Kit, Omega Bio-Tek, GA) and PCR amplified the full-length coding sequence of the putative TbLC2 gene (Tb927.9.12820, see Results) from the genomic DNA (Table A in S1 Text for primers). We also PCR amplified His6/eGFP (for nickel column affinity and fluorescent localization) and the biotin-binding domain of *Chlamydomonas*'s biotin carboxyl carrier protein (BCCP) [80, 81] coding sequences (Table A in S1 Text for primers) from the pOCC98 plasmid [82] (gift from Aliona Bogdanova, Max Planck Institute of Molecular Cell Biology and Genetics) and the wt-lc2-bccp plasmid [80] (gift from Ritsu Kamiya, Kyoto University, Kyoto Japan), respectively. We designed all three sets of primers with the NEBuilder Assembly Tool (https://nebuilder. neb.com/) and assembled the PCR products using the HiFi DNA assembly one-step multiple fragment cloning technique (NEBuilder HiFi DNA assembly cloning kit, E5520, New England Biolabs, MA). We ligated the assembled products into a pXS2 plasmid [83] using the pXS2. Pex13.2 [84] plasmid (gift from Meredith Morris, Clemson University, SC) from which we excised the Pex13.2 fragment with ClaI (R0197S, New England Biolabs, MA) and EcoRI restriction enzymes (R3101S, New England Biolabs, MA). We used the newly constructed pXS2.TbLC2.BCCP.His6/eGFP plasmid only as a convenient intermediate vector from which to PCR amplify the assembled TbLC2.BCCP.His6/eGFP sequence using TbLC2.BCCP.His6/ eGFP Forward and TbLC2.BCCP.HIS6/eGFP Reverse primer set (Table A in S1 Text for primers).

We used pLEW100V5 [79, 85] (a gift from George Cross, Rockefeller University; Addgene plasmid # 24011; http://n2t.net/addgene:24011; RRID:Addgene_24011), a pLEW100 vector derivative, to create an inducible expression vector system with phleomycin resistance, which enabled selection of stable transfectants. The pLEW100V5 plasmid has T7 terminators that regulate the rRNA promoter and drives the extremely high gene expression level. Overexpression of the recombinant TbLC2 facilitated the stable incorporation of the TbLC2.BCCP.His6/ eGFP (called TbLC2::eGFP in the Results and Discussion sections) into the axoneme, enabling it to out-compete any endogenous TbLC2 proteins. We excised the luciferase gene and linearized the pLEW100V5 plasmid with HindIII (R3104S, New England Biolabs, MA) and BamHI (R3136S, New England Biolabs, MA) restriction enzymes. We then assembled the PCR amplified TbLC2.BCCP.His6/eGFP dsDNA segment into the linearized pLEW100V5 vector using HiFi DNA assembly to generate the pLEW.TbLC2.BCCP.His6/eGFP plasmid (S13 Fig). The assembly of coding sequences of TbLC2, BCCP, eGFP, and His6 into a single continuous open reading frame ensured the expression of a ~50 kDa protein structure once induced in trypanosome cells. We verified the correct insertion of LC2::BCCP His6/eGFP into the linearized pLEW100V5 plasmid by agarose gel electrophoresis and Sanger sequencing using the pLEW sequencing primers set (Table A in S1 Text for primers) [86].

While designing the vectors and cloning strategy, we ensured the vectors had only one unique NotI restriction site later used for linearization and stable integration into the rRNA locus [87].

## Cloning RNAi constructs

To knock down the expression of targeted genes in trypanosomes, we used the tetracycline-inducible pZJM RNAi vector [88]. The pZJM vector contains a phleomycin selection marker and a sequence to facilitate the homologous integration of the linearized vector at the silent rRNA spacer locus [88].

To generate the FLAM3 RNAi knockdown plasmid pZJM.FLAM3 (S13 Fig), we excised the tubulin gene from a pZJM.tubulin RNAi plasmid (a gift from James Morris, Clemson University) using XhoI (R0146S, New England Biolabs, MA) and HindIII (R3104S, New England

Biolabs, MA) restriction enzymes and inserted a 383 bp fragment of FLAM3 (Tb927.8.4780 [33]) that we PCR amplified (Table A in S1 Text for primers) from the *T. brucei* genomic DNA into the linearized RNAi vector.

To generate the TbLC2 RNAi knockdown plasmid, pZJM.TbLC2 (S13 Fig), we excised the FLAM3 fragment out of pZJM.FLAM3 using XhoI and HindIII restriction enzymes and ligated in a 125 bp fragment of 3'-UTR region of TbLC2 that we PCR amplified from extracted genomic DNA using TbLC2 3'-UTR XH primer set (Table A in S1 Text for primers). We targeted the 3'-UTR region of TbLC2 to ensure that the RNAi only knocks down the endogenous gene and not the expression of recombinant TbLC2::eGFP in LC2 overexpression cell lines as the recombinant TbLC2 does not contain its UTRs.

To generate the FLAM3/LC2 double RNAi knockdown plasmid, pZJM.FLAM3.TbLC2 (S13 Fig), we linearized the pZJM.FLAM3 plasmid with the XhoI restriction enzyme and ligated in the LC2 3'-UTR target sequence that we PCR amplified from genomic DNA using TbLC2 3'-UTR XX primer set (Table A in S1 Text for primers). We used calf intestinal alkaline phosphatase (M0290, New England Biolabs, MA) to dephosphorylate the 5' and 3' ends of the linearized plasmid to prevent self-ligation of linearized plasmid [89]. By including the RNAi target for both FLAM3 and LC2 in the same plasmid, under the same promotor, we created double knockdown cell lines without performing a double transfection and risking losing one or the other plasmid or requiring another selection marker.

## Maintaining and culturing *T. brucei* cells

We used the procyclic form of *Trypanosoma brucei brucei* (strain 29–13, a gift from James Morris, Clemson University, hereafter called wild-type) for all the genetic modification and epitope tagging. *T. brucei* 29–13 was derived from the 427 strain [90], and it encodes a T7 polymerase (selected for using G418, geneticin, at a final concentration of 15 μg/mL) and a tetracycline repressor (selected for using hygromycin at a final concentration of 50 μg/mL) for induced gene expression [90]. We grew the cells on growth media consisting of SDM-79 [91] (custom produced by Life Sciences Research Products, Thermo Fisher Scientific for the Eukaryotic Pathogens Innovation Center labs, Clemson University, Clemson, SC) supplemented with 10% heat-inactivated tetracycline-free fetal bovine serum (S162TA, Biowest, MO) and porcine hemin at 7.5 μg/mL final concentration (AAA11165-03, Alfa Aesar, MA) to have a population doubling time of 8–9 hours with a logarithmic growth phase between $1 \times 10^6$ and $1 \times 10^7$ cells/mL [92]. We maintained the cells in a humidified environment supplied with 5% $CO_2$ and at 27°C by diluting the culture 10-100-fold every 2–4 days into 5 mL of growth media in 25 cm$^2$, 50 mL vented cap culture flasks. We induced knockdown and overexpression in cells transfected with tetracycline-inducible plasmids pZJM and pLEW100V5 by dosing the cells with 1 μg/mL final concentration of doxycycline.

To assess the effect of physical shaking on cell growth, clustering, and clumping, we split the doxycycline-induced cell cultures into two flasks. We placed one on an orbital shaker (Advanced Dura-shaker, 10159–960, VWR International, LLC) set at 85 rpm within the incubator and cultured the other one under normal conditions, without shaking. We chose these shaking conditions to approximate the shear stresses induced on swimming trypanosome cells in cell lines with RNAi knockdown-based motility defects (S1 Text for an order of magnitude estimate on the shear stresses on trypanosome cells).

To monitor the growth of trypanosome cell cultures, we briefly agitated the culture flask to disperse the cells, took a small aliquot with a pipette, and used a cell counting chamber (Neubauer-improved, Paul Marienfeld GmbH & Co. KG, Lauda-Königshofen, Germany) to count the cells and quantify the cell density.

## Transfecting *T. brucei* cells

We transfected plasmids into *T. brucei* cells using electroporation [93] to generate the cell lines used in this study (Table B in S1 Text for cell lines). We maintained the cells at mid log phase ($5$–$10 \times 10^6$ cells/mL) for at least two generations to optimize transfection efficiency. We used a total of $5 \times 10^7$ cells for each transfection and always mirrored the transfection with negative control (with just water instead of DNA) to monitor the action of selection drugs. In short, we linearized 40 µg of the vector with NotI (R0189S, New England Biolabs, MA), mixed the linearized plasmid with $5 \times 10^7$ cells that we had washed with 10 mL of cytomix (120 mM KCl, 25 mM HEPES, 0.15 mM $CaCl_2$, 10 mM $K_2HPO_4$, 2 mM EDTA, 5 mM $MgCl_2$, titrated to pH 7.6 with KOH) [94] and resuspended to 400 µL into a 4 mm cuvette. We electroporated the cells with two pulses (exponential decay pulse mode, 1500 V, 24 µF) separated by 10 s (Gene Pulser Xcell Electroporation System, Bio-Rad Laboratories, Inc., CA). We added 12 mL of growth media supplemented with 15 µg/mL G418 and 50 µg/mL hygromycin and put 0.5 mL of the electroporated cells per well into a 24-well plate. After incubation for 24 hours, we added 0.5 mL of growth media supplemented with selection drugs (20 µg/mL of blasticidin for pLEW100V5 and 5 µg/mL of phleomycin for pZJM vectors, final concentration) to each well and allowed the cells to grow for 10–14 days at 27˚C and 5% $CO_2$ and checking the wells every other day for cell growth.

We used a limiting dilution method [95] to generate a clonal cell population from the pool of stable cells by seeding 0.5 cells per well into a 24-well plate containing 400 µL of conditioned growth media (prepared by growing wild-type cells to a density of ~$10^7$ cells/mL in growth media to mimic the environment of a well-grown colony for the growth of single cell in a well) supplemented with the corresponding selection drugs. We induced the clonal cell populations from each well and screened for eGFP expression using flow cytometry, detached cilium using bright field microscopy for RNAi knockdowns, and quantitative reverse transcription PCR (RT-qPCR) for FLAM3 and LC2 knockdowns.

## Flow cytometry

We assessed the fraction of the fluorescent cells in the total cell population, as well as their eGFP expression level, using flow cytometry (CytoFLEX LX, Beckman Coulter, CA, USA). We harvested 1 mL of mid-log phase ($5$–$10 \times 10^6$ cells/mL) wild-type and transgene-induced cells by centrifuging them at 1000×g for 10 minutes. We washed the cell pellets with 1 mL of PBS and resuspended them in 2 mL PBS. We analyzed the GFP fluorescence of ~20,000 cells using the FITC channel and compared the eGFP fluorescence of cells within an experiment, but not between the experiments conducted at different times.

## Confirmation that RNAi suppresses the expression of TbLC2 using RT-qPCR

The morphological phenotype of the cilium separating from the cell body (Fig 2) simplified our assessment of the FLAM3 RNAi knockdown effectiveness. However, we did not observe morphological differences between FLAM3 KD and FLAM3-LC2 KD cells with light (Fig 2C) or scanning electron (Fig 2B) microscopy, leaving open the question of whether the LC2 knockdown was effective. However, our RNAi double knockdown strategy of putting both RNAi targets under the same promotor (S13 Fig) and the extensive detachment of the cilium from the cell body from both the FLAM3 KD and FLAM3-LC2 KD cells (Fig 2) suggest that the LC2 RNAi was likely successful, as well.

We confirmed the extent of FLAM3 and endogenous TbLC2 RNAi knockdown using semi-quantitative reverse transcription PCR (RT-qPCR). We isolated the total RNA from $2\times10^7$ cells using a Monarch Total RNA Miniprep Kit (T2010, New England Biolabs, MA) as described by the manufacturer with an additional 30-minute, room temperature, in-tube DNAse I (M0303S, New England Biolabs, MA) treatment to eliminate the gDNA contamination. We used the Luna Universal One-step RT-qPCR kit (E3005S, New England Biolabs, MA) to determine the relative abundance of target mRNA between two samples. We prepared 20 μL samples normalized by total RNA content and performed the reverse transcription and cDNA amplification (gene-specific RT-qPCR primer sets, Table A in S1 Text) in a single reaction using a CFX Opus real-time qPCR machine (CFX96, Bio-Rad Laboratories, CA). Briefly, we used a single reverse transcription step (10 min at 55˚C) and an initial denaturation (1 min at 95˚C), followed by 40 cycles of 15 s denaturation and 30 s extension.

We performed semiquantitative RT-qPCR analysis of the threshold cycle, Ct, data to calculate fold change in target mRNA expression, as described previously [41, 42]. We normalized the reactions with total RNA content and compared the cycle threshold between the wild-type and LC2 KD and FLAM3-LC2 KD cells.

## *T. brucei* sedimentation assay

We quantified the motility of *T. brucei* cells by examining their sedimentation rate. Since cells with motility defects tend to settle to the bottom of an undisturbed culture, we quantified the extent of motility defects by measuring the relative decrease in the absorbance at 600 nm (OD600) near the top of a cuvette with cells [96]. We put 500 mL of log phase density ($1\times10^6$–$1\times10^7$ cells/mL) cells into 1 mL disposable cuvettes (UVette, 952010051, Eppendorf AG, Hamburg, Germany). We measured the OD600 every hour for 7 hours in each of two cuvettes for each cell line, with one cuvette mixed by pipetting at each measurement and the other kept undisturbed throughout the experiment, using a spectrophotometer (V-1200 Spectrophotometer, 10037–434 VWR International, LLC). To isolate the effect of motility defects from the change in OD600 due to cell growth defects, we calculated the ΔOD600 by subtracting OD600 reading for mixed cuvette from that for stationary cuvette. We performed all sedimentation assays in triplicate.

## Immunofluorescence microscopy

We examined the subcellular localization of eGFP tagged TbLC2 using wide-field fluorescence microscopy. We took widefield immunofluorescence images using an inverted microscope (Eclipse Ti-E, Nikon Instruments, Inc.) with a Plan Apo 60x water immersion objective. We acquired the images at 16-bit depth and 200–2000 ms exposure time with a CCD camera (CoolSNAP HQ2 Monochrome, Photometrics, AZ). We took confocal immunofluorescence images using an upright microscope (TCS SPE-II with DM250 RGB, Leica Microscopes, Buffalo Grove, IL) with a 63x objective. We processed the images and performed quantitative analysis using ImageJ [97, 98].

We cultured eGFP expressing cell lines in induction media to a density of $5\times10^6$ cells/mL. We washed the cells with PBS, fixed them on a glass slide using 2% paraformaldehyde, and rinsed them with ample wash solution (0.1% normal goat serum, NGS, in PBS) before applying a 0.1 M glycine solution to quench unreacted aldehydes after fixation and hence reduce background fluorescence caused by aldehydes [99]. After 15 minutes, we rinsed the cells with wash solution and permeabilized them with permeabilization solution (0.5% Triton-x 100 in PBS). We washed the cells with wash solution and added 50 μl of mounting solution (DAPI

Fluoromount-G, 0100–20, Southern Biotech, Birmingham, AL). Finally, we sealed the sample with a coverslip and nail polish.

We probed the localization of paraflagellar rod (PFR) and the tubulin-rich axoneme in permeabilized, fixed the cells using indirect immunofluorescence tagging [100]. Briefly, we washed the cells with ample wash solution and blocked the sample using blocking solution (10% NGS and 0.1% Triton X-100 in PBS) for 1 hour at room temperature. We incubated the cells with either an anti-PFR2 antibody [10] (rabbit anti-PFR2 serum, a gift from Kent Hill, UCLA) or the DM1A monoclonal anti-α-tubulin antibody raised in mice (MBAT205, MilliporeSigma, Burlington, MA). After washing the sample with ample wash solution, we incubated the anti-PFR2 antibody-stained cells with an Alexa Fluor 635 conjugated goat anti-rabbit secondary antibody (A-31577, Thermo Fisher Scientific, Inc., MA) diluted 100-fold and the DM1A-stained cells with an Alexa Fluor 594 conjugated goat anti-mouse secondary antibody (A-11005, Invitrogen, Waltham, MA) in blocking solution for an hour. Subsequently, we washed and mounted cells with DAPI-containing mounting solution and finally sealed them with a coverslip as described above.

## Extraction of cilia

We harvested $1\times10^8$ FLAM3 RNAi-induced cells by centrifuging at 1500×g for 10 min at 4°C, washed the pellets in PEME buffer (EGTA 2 mM, MgSO$_4$ 1 mM, EDTA 0.1 mM, PIPES free acid 0.1 mM, pH 6.9), and resuspend them into a 50 mL conical tube with 2 mL of PEME. We vortexed the cells for 15 minutes at 3200 rpm and centrifuged them at 420×g in a swinging bucket rotor for 10 minutes. We separated the dissociated cilia from the cell bodies by transferring the supernatant to 1.5 mL microcentrifuge tubes and centrifuging them at 420×g for 5 minutes. We saved the supernatant (cilia extract) and resuspended all the pellets in a 50 mL conical tube with 2 mL of PEME buffer. We performed a second separation by repeating the centrifugation steps. We combined the supernatant-containing ciliary extracts and centrifuged them at 25,000×g for 20 minutes. We stored the cell body and ciliary pellets at -80°C.

We modified a previously published method of cell fractionation to extract the cilia [48] from FLAM3 uninduced or wild-type cells with cilia still intact to the cell body. Briefly, we first harvested and washed $5\times10^7$ cells with PEME buffer as described above. We then detergent extracted the cells by resuspending the cell pellet in PEME buffer with 1% IGEPAL (CA-630, Alfa Aesar, MA) and incubating on ice for 15 minutes. We centrifuged the cells at 3400×g for 6 minutes at 4°C and saved the membrane and cytoplasmic content-containing supernatant (S1) and the cilia and subpellicular microtubule corset-containing pellet (P1). We resuspended P1 in PEME buffer with 1% IGEPAL and 1 M NaCl and incubated the mixture for 45 minutes on ice to depolymerize the corset microtubules. We then centrifuged the mixture at 16000×g for 15 minutes at 4°C to separate it into a depolymerized corset microtubule and associated protein-containing supernatant (S2) and an axoneme, paraflagellar rod, and basal body-containing pellet (P2).

## Western blotting

We probed the expression of eGFP tagged TbLC2 in various cell and ciliary fractions using anti-eGFP antibodies and standard immunoblotting methods. Briefly, we generated cell lysates from various cellular and ciliary fractions, incubated them in SDS-sample buffer at 95°C, loaded them into polyacrylamide gel wells and ran them for approximately 45 minutes at 190 V in a mini-PROTEAN tetra vertical electrophoresis cell (Bio-Rad Laboratories, Inc). We transferred the protein bands onto PVDF membranes (170–4156, Bio-Rad Laboratories, Inc) with a Trans-Bolt Turbo System (Bio-Rad Laboratories, Inc), blocked the membrane in

blocking buffer (1% skimmed milk powder prepared in PBS with 0.1% Tween), incubated it in mouse anti-GFP primary antibody (SC-9996, Santa Cruz Biotechnology, Inc.) diluted to 1:800 in blocking buffer, washed the membrane thoroughly with PBST (0.1% Tween in PBS), and incubated it in alkaline phosphatase-conjugated goat anti-mouse secondary antibody (31328, ThermoFisher Scientific) diluted to 1:5000 in blocking buffer. We detected the bound alkaline phosphatase labeled antibodies by incubating the membrane in BCIP/NBT (5-bromo, 4-chloro, 3-indolylphosphate/nitro-blue tetrazolium, AMRESCO, LLC.) substrate for 30–45 minutes. Finally, we scanned the membrane (Epson Perfection V700 photo scanner) and quantified the bands with ImageJ [97, 98]. We also probed the samples using rabbit anti-detyrosinated tubulin primary antibody (AB3201, Sigma-Aldrich, MO) diluted to 1:5000 in blocking buffer and alkaline phosphatase-conjugated goat anti-rabbit (31342, Thermo Scientific, MA) secondary antibody diluted 1:7500 in blocking buffer as a loading control.

## Electron microscopy

We imaged the structure of the cilia with negatively stained electron microscopy. We fixed the cells by incubating 5 mL of culture with electron microscopy grade glutaraldehyde (16300, Electron Microscopy Sciences, Inc., PA) at a final concentration of 2.5% at room temperature for 10 minutes followed by and an overnight fixation with Karnovsky's fixative reagent (2.5% glutaraldehyde, 2% paraformaldehyde, and 0.1% tannic acid prepared in 0.1 M phosphate buffer, pH 7, 15720, Electron Microscopy Sciences, Inc., PA). We washed the cells once with 0.1 M pH 7.2 phosphate buffer, resuspended them in 0.1 M pH 7.2 phosphate buffer with 0.02% sodium azide, and stained them with 1% osmium tetroxide before running the sample through a graded ethanol series to dehydrate the samples. For TEM, we infiltrated the ethanol-soaked samples with LR white embedding resin and cured them for 24 hrs in a 60˚C oven. We sectioned the samples with a microtome, placed thin sections on copper grids, and captured the transmission electron brightfield micrographs with a Hitachi SU9000 UHR operated at 100 kV. For SEM, we soaked the dehydrated sample in 1:1 ethanol:HMDS (hexamethyldisilazane, 16700, Electron Microscopy Sciences, Inc., PA), followed by pure HMDS. We air-dried the samples overnight, placed them onto silicon wafers, and sputter coated them with platinum. Once dry, we imaged the samples using SEM (Hitachi SU5000 at an accelerating voltage of 2kV).

## Motility traces

We tracked the motility of trypanosome cells using widefield light microscopy and quantitative image analysis. We prepared a 100–150 μm deep motility chamber constructed from parafilm (PM-999, Bemis, WI) sandwiched between easy cleaned [101] glass slides and No. 1.5 22 mm x 22 mm cover glasses (16004–302, VWR). We passivated the chamber by incubating it with 0.25% poly-L-glutamate (P4886, Sigma-Aldrich, Inc.) [22] prepared in PBS for 25 minutes and washing out the excess poly-L-glutamate with distilled-deionized water and ethanol. We diluted cultured and induced cells to a final density of $1 \times 10^6$ cells/mL using fresh growth media and equilibrated them to room temperature. We transferred 10 μL into the motility chamber, sealed it with the nail polish, and imaged it with phase-contrast microscopy (Aus Jena Telaval 3 inverted binocular microscope) at 5x magnification (Aus Jena planachromat, NA 0.1). We recorded movies at 45 frames per second using a CCD camera (Grasshopper3 USB3 GS3-U3-15S5M-C, Teledyne FLIR, LLC.) for 10 seconds.

We preprocessed the image sequences in ImageJ [97, 98, 102] by subtracting the maximum projection from each frame. We thresholded the images and applied a spot-enhancing 2D filter

(SpotTracker plugin [103, 104]) to further enhance the cell spot and reduce the background noise. We cropped the frames to include only single cells. We quantified motility by tracking single, non-intersecting cell paths to minimize the breaking of tracks using the wrMTrck [105] and ParticleTracker [106] plugins with the following parameters: a) minSize of 100 pixels$^2$, b) maxSize of 400 pixels$^2$, and c) threshMode Otsu for wrMTrck and a radius of 15 pixels and cutoff of 3 pixels for ParticleTracker. We tracked cells over the entire sequence of frames (10 s) to minimize bias due to inequality in tracked time intervals [57].

We used DiPer [57] to calculate the directional ratio, average swimming speed, and average velocity of the tracked trypanosomes. Briefly, directional ratio (DR) is the magnitude of the total vector displacement (final position minus initial position, $|\Delta\vec{r}| = |\vec{r}_2 - \vec{r}_1|$) divided by the scalar distance (integrated path length, $L$) of the cell along its track ($DR = \frac{|\Delta\vec{r}|}{L}$). We used the DR calculated at the last time point of each track to classify the cell population into three groups: A. high persistence (DR > 0.2, most persistent swimmers), B. medium persistence (0.05 < DR < 0.2, persistent swimmers with intermittent tumbling), and C. low persistence (DR < 0.05, tumblers) (S3, S4 and S5 Movies for examples of each). Average swimming speed is the integrated path length, $L$, of the cell along its track divided by the time it took to traverse that distance ($s = \frac{L}{\Delta t}$). Average velocity is the total vector displacement, $\Delta\vec{r}$, divided by the time it took to displace ($\vec{v} = \frac{\Delta\vec{r}}{\Delta t}$).

## Optical tweezer assay and power spectral density analysis

We used a custom-built, single-beam optical tweezer to trap the cells and measure the fluctuation of the transmitted laser intensity in the back focal plane of the condenser due to the scattering of the laser caused by cell displacements driven by rotation and ciliary beating. Optical tweezer-based frequency analysis has multiple advantages over the quantitative image analyses of high-speed video microscopy alternatives, including fewer constraints on cell rotations (S1 and S2 Movies) and out-of-plane beating waveforms than occur in thin (<10 μm) imaging chambers used for microscopy, the ability to probe higher order frequencies and get better resolution with power spectral density analysis of data collected at 10 s of kHz, and higher throughput than having to perform time-consuming video analyses. Briefly, the optical tweezers use a 1064 nm, 10 W ytterbium fiber laser (YLR-10-1064-LP, IPG Photonics) to generate the trapping laser beam focused into the sample plane using a CFI Plan Apochromat Lambda 60x N.A. 1.4 oil immersion objective lens (Nikon Instruments, Inc.).

We harvested cultured and induced cells in the log growth phase and diluted them to a final concentration of $1\times10^6$ cells/mL in fresh growth media. We flowed the cells into a 2–3 mm wide chamber constructed as described above, sealed the chamber with nail polish, and allowed the media to equilibrate to room temperature for 10–15 minutes. We trapped the cells at minimal laser power at the sample plane (10–25 mW) to maintain cell viability by minimizing photo and thermal damage [50]. We captured the time series laser fluctuation data with a quadrant photodiode (QPD, QP45-Q HVSD, First Sensor Inc.) using back focal plane detection [107] at 500 kHz and smoothened by averaging to a final rate of 25 kHz using an FPGA (PXI-7854R, NI) and custom-written LabView (NI) VIs. We calculated the power spectral density (PSD) from the time series data using the FFT Power Spectrum and PSD built-in LabView functions and averaged three such PSDs for each measurement. We used the QPD sum signal fluctuation (represents position fluctuation axially [108]) and difference signal fluctuation (represents position fluctuation in the x-y plane [108]) for determining the ciliary beat and cell rotation frequencies, respectively, in the PSD plot.

## High-speed imaging, shape tracking, and measuring curvature and ciliary beat switch

We recorded high-speed image sequences in an assay chamber constructed using microscopic beads as spacers as previously described [101], with minor modifications. In brief, we 10-fold diluted a 5% w/v stock of polystyrene beads (PP-50-10, Spherotech, Inc.) and put 5 μm onto an easy cleaned [101] microscope glass slide. We placed an 18 mm x 18 mm easy cleaned coverslip on the bead solution droplet and sealed the sides of the coverslip (perpendicular to the length of the slide) with nail polish. We washed the beads out with distilled deionized water using vacuum suction. The chambers had a depth of 5.65 ± 0.12 μm (as measured by focusing the objective at the two inner surfaces of the chamber, mean ± SEM, N = 16), which was larger than the cell width (1.5–3.5 μm [109]) but smaller than the cell length (S6 Fig) and thus confined the cell body to the focal plane while allowing for free ciliary beating.

We harvested cultured and induced cells in the log growth phase and diluted them to a final concentration of $1 \times 10^6$ cells/mL in fresh growth media. We sealed the chamber with nail polish and immediately imaged the cells with bright field illumination using a CFI Plan Apochromat Lambda 60x N.A. 1.4 oil immersion objective lens (Nikon Instruments, Inc.). We captured 3-second image sequences at 200 frames per second with a minimum exposure time (250 μs, to minimize motion blur) using a high-speed CMOS camera (M-PRI-1000, AOS Technologies AG). We slightly defocused the sample stage to facilitate the shape tracking.

As explained in the motility tracking section above, we preprocessed the image sequences in ImageJ [97, 98]. We modified custom-written MATLAB scripts [110] (The MathWorks, Inc.) to track individual beating cilia and calculate the tangent angle with respect to a horizontal reference axis, $\psi$, at equally spaced points along the ciliary arc length, $s$ (Veikko Geyer and Benjamin Friedrich provided the source code for both scripts).

To determine the ciliary beat amplitude (A), we first subtracted the average ciliary shape from the tangent angles measured at each point along s and calculated the amplitude as the half-width of the waveform shape formed by the tangent angle distributions [23]. We then measured the ciliary beat wavelength as twice the separation, in terms of arc length normalized to the contour length (s/L), between the maximum and the minimum tangent angles. We calculated the curvature as a function of s/L, $\kappa\left(\frac{s}{L}\right) = \frac{d\psi}{d\left(\frac{s}{L}\right)}$. We calculated the curvature at $\frac{s}{L} = 0.1$, as measured from the cilium's tip, and used the absolute values of the curvature (to prevent the oppositely directed curvatures canceling each other) to calculate the time average curvature-at-the-tip.

We calculated the maximum curvature asymmetry ratio, $AR_{MC}$, by dividing the maximum curvature found in each opposing bend direction as a single ciliary bend wave propagated along the cilium. Since we have no marker for cell or cilium orientation, we calculated the asymmetry ratio such that $AR_{MC} \geq 1$. An $AR_{MC} = 1$ indicates that the bending wave propagated symmetrically along the cilium, and an $AR_{MC} > 1$ indicates that the bending wave propagated asymmetrically along the cilium.To determine the fraction of time the cells spend in the tip-to-base (regular) and base-to-tip (reverse) ciliary beating modes, we analyzed the ciliary beating of each cell, frame by frame, over the entire period of 3 seconds (600 frames). We noted the reverse beat dwell time as the interval between the initiation of base-to-tip ciliary bend propagation (identified by the traveling of the ciliary bend initiated at the ciliary base towards the tip over the 600 frames) to the initiation of the next regular ciliary bend at the tip and divided the dwell time by 3 seconds (or 600 frames).

## Supporting information

**S1 Text. Supporting discussion and supporting tables A and B.**
(PDF)

**S1 Fig. Multiple sequence alignment of putative Tctex-type dynein light chains in the *Trypanosoma brucei* genome and LC2 homologs from other genomes.** *T. brucei* A and B represent the two closest hits for the *C. reinhardtii* homolog of LC2 (Tb927.9.12820 and Tb927.11.7740, respectively). The color bands indicate the level of sequence conservation, with red indicating identical residues, and the spectrum between orange and yellow indicates level of sequence conservation from high to low. White residues are not conserved. (Materials and methods).
(EPS)

**S2 Fig. Sequence alignment of TbLC2 (*T. brucei* A) and the *C. reinhardtii* homolog of LC2 (*C. reinhardtii*).** Conservation of hydrophobicity (*red*) and polarity (*blue*) are indicated. The spectrum from light red to dark red indicates lower to higher hydrophobicity, and the spectrum from purple to blue indicates low to high polarity (charge). The *C. reinhardtii* LC2 residues above the black lines are in the LC9 binding interface as determined by using the change in the accessible surface area (dASA) in PyMol (PyMol 2.3.4, Schrodinger, LLC).
(EPS)

**S3 Fig. Multi-histogram plot of eGFP fluorescence intensity measured by using flow cytometry for TbLC2::eGFP overexpressed cell lines.** The frequencies of occurrence are normalized to the mode to facilitate comparisons across cell lines with different cell counts registered by the flow cytometer. The wild-type cell line was used as a control. Cell counts are N = 517500, 61144, and 130600 for WT, FLAM3-LC2 KD/LC2 OE, and WT/LC2 OE cells, respectively.
(EPS)

**S4 Fig. mRNA expression level of FLAM3 and LC2 genes in LC2 KD and FLAM3 KD/ LC2 KD cell lines determined by semiquantitative RT-qPCR.** The percentage reported is relative to the expression of the same genes in the WT cells. The error bars represent SEM calculated from triplicates of the experiment performed on the total RNA sample purified.
(EPS)

**S5 Fig. Western blot analysis of LC2::eGFP.** Anti-GPF antibodies stained LC2::eGFP in uninduced (-dox) and induced (+dox) whole-cell (WC), cell body only (CB), and ciliary (cilia) fractions. We used tubulin as a loading control. We extracted the cilia using mechanical shearing and loaded an equal number of cells in all lanes except in the ciliary fractions, which contains cilia extracted from 15-fold as many cells as used in the other lanes. We used tubulin (bottom) as a loading control.
(EPS)

**S6 Fig. FLAM3 knockdown and TbLC2 overexpression cause shorter *T. brucei* cilia.** Length distributions of cilia from strains as indicated. The black lines indicate the mean length, and shaded regions around the line indicate the SE of the mean (N = 76, 76, 52, and 101 for WT, FLAM3 KD, FLAM3-LC2 KD, and FLAM3-LC2 KD/LC2 OE cells, respectively). *** represents *p*-value < 0.0001 and ** represents *p*-value < 0.001.
(EPS)

**S7 Fig. LC2 knockdown causes mislocalization of kinetoplast and cell division defects. A**. Representative images of uninduced, FLAM3 KD, FLAM3-LC2 KD, and FLAM3-LC2 KD/ LC2 OE cells cultures in the culture flask using phase-contrast microscopy 72 hours post-

induction when we did not shake (*top*) and shook (*bottom*) the flasks. Major clusters of cells are indicated (red arrows). The scale bars represent 10 μm. **B**. Representative DAPI stained images for classification of cells as having x kinetoplasts (xK) and y nuclei (yN). 1K 1N refers to cells with one kinetoplast normally localized to one nucleus (*left*). MK MN refers to cells classified as having multiple (M>2) mislocalized (closer to each other) kinetoplasts and nuclei (*right*), likely resulting from incomplete kinetoplast migration and/or incomplete cytokinesis. The scale bar represents 5 μm and both images in this panel have the same scale. **C**. Occurrence frequency of one kinetoplast and one nucleus, normally localized within the cell (1K 1N) and the occurrence frequency of the multi-kinetoplast, multi-nucleus (MK MN) classification, as described in panel **B**., in uninduced and induced (72 hours post-induction) LC2 KD, FLAM3 KD, FLAM3-LC2 KD, FLAM3-LC2 KD/LC2 OE, and WT/LC2 OE cells. N = 101, 192, 122, 111, 72, and 70 total classified cells of each strain, respectively. Other classifications, e.g., 1K 2N, 2K 1N, and 2K 2N, which likely include cells undergoing cell division, account for the percentages not represented. The error bars represent the statistical counting error. ** = p-value < 0.001 and *** = p-value < 0.0001, two-tailed paired t-tests. **D**. DIC microscopy image of fixed FLAM3-LC2 KD cells, including a representative amorphous clump of cells with multiple detached cilia (*red arrow*).
(TIF)

**S8 Fig. Shaking does not rescue the cell growth defects in LC2 knockdown cells.** Growth curves for FLAM3-LC2 KD (*red*) and FLAM3-LC2 KD/LC2 OE (*orange*) cell lines, comparing the effect of mechanical agitation. Shaken (*dashed lines*) and not shaken (*solid lines*) indicate incubation of cell culture with or without orbital shaking, respectively at 80–90 rpm for the duration of incubation. Points represent the mean of three cultures, and error bars represent the standard error of the mean.
(EPS)

**S9 Fig. Tangent angle characterization of the ciliary beat waveform in FLAM3 knockdown cells.** Example tangent angle (ψ(s/L), *inset*) plotted as a function of arc length (s) normalized to the contour length (L) along the cilium for a typical tip-to-base wave propagation in FLAM3 KD cells (*left*), and the same tangent angle plotted after subtraction of the average ciliary beating shape (*right*). Data from an example frame (*green*) used to calculate the beat amplitude and wavelength (*black arrows*) and the average shape (the static component of the waveform, *red*) are highlighted both before (*left*) and after (*right*) the average shape subtraction. This cilium exhibits only slightly non-zero mean curvature because the average shape of the ciliary beating waveform has only a slightly non-linear time-averaged tangent angle (*red*, *left*). This cilium exhibits an approximately symmetric dynamic curvature because the time-averaged tangent angle subtracted waveform (e.g., *green*, *right*) has approximately equal positive and negative maximum curvature. s/L of zero represents the ciliary base in both plots.
(EPS)

**S10 Fig. Curvature during the highly asymmetric reversed base-to-tip beating in FLAM3/ LC2 double knockdown cell.** The curvature normalized to contour length (κ/L) plotted as a function of normalized arc length (s/L) along the cilium for a typical highly asymmetric base-to-tip wave propagation in FLAM3-LC2 KD cells. s/L of zero represents the ciliary base, and the colors (*red* to *magenta*) represent time progression during the propagation of a ciliary bend in a single ciliary beat period. The t = 0 beat profile (*red*) represents ciliary bend initiation towards the base (left side), followed by the successive profiles representing the propagation of the bend towards the tip (*right side*).
(EPS)

**S11 Fig. The static component of FLAM3-LC2 KD cells ciliary beating waveform.** The mean tangent angle was calculated as a function of normalized arch length (s/L) from the average shape of the beating waveform for forward (tip-to-base) and reverse (base-to-tip) beating waveforms of FLAM3-LC2 KD cells. Each point represents the mean of the average tangent angle from 8 cells, and the error bars represent the standard error of the mean.
(EPS)

**S12 Fig. Full-length TbLC2 has a high degree of structural conservation with the N-terminal truncated TbLC2 protein, as calculated by AlphaFold.** The template-free model of TbLC2 with the 22 N-terminal amino acids truncated (*left*) to match the resolved residues in the cryo-EM (Fig 1B), and the template-free model of full-length TbLC2 (*right*), both as calculated by AlphaFold.
(EPS)

**S13 Fig. Vector maps of overexpression and RNAi plasmids used.** Vector maps of the tet-inducible pLEW.TbLC2::BCCP His6/eGFP plasmid generated by cloning the preassembled TbLC2, BCCP, His6, and eGFP genes into the ORF of pLEW100V5 plasmid by excising the luciferase gene (*gray*). The expression of the TbLC2::BCCP His6/eGFP gene is driven by the strong rRNA promoter, whereas the T7 promoter drives the expression of the selection marker (blasticidin). The rRNA spacer acts as a target locus during transfection. Vector map of the pZJM.FLAM3 RNAi vector generated by replacing the tubulin sequence (*gray*) between XhoI and HindIII restriction sites with the FLAM3 sequence. Vector map of the pZJM.TbLC2 RNAi vector generated by replacing the FLAM3 (*gray*) sequence between XhoI and HindIII restriction sites with TbLC2 3'-UTR. Vector map of the pZJM.FLAM3.TbLC2 RNAi vector generated by inserting the TbLC2 3'-UTR sequence at the XhoI restriction site. All the RNAi vectors are integrated into the trypanosome genome at the rDNA site.
(EPS)

**S14 Fig. LC2 KD and WT/LC2 OE do not show cell clustering defects.** Representative images of LC2 KD and WT/LC2 OE cell in the culture flask using phase-contrast microscopy 72 hours post-induction when we did not shake the flasks. We observed no cell clustering. The scale bars represent 10 μm.
(EPS)

**S1 Movie. Trapped cells displayed unconstrained-like ciliary beating behavior.** Movie of an uninduced cell trapped by the optical tweezer 70–100 μm above the surface of the motility chamber. The cell is free to rotate and maintain the out-of-plane nature of its beating waveform. The power spectra used to quantify the beat frequency in Fig 6 were obtained with cells trapped 70–100 μm above the surface of the surface. Movie recorded using wide-field microscopy with a 60x objective at 45 fps and played back at the same frame rate.
(AVI)

**S2 Movie. Trapped cells near motility chamber surface displayed planar ciliary beating.** Movie of an uninduced cell trapped by the optical tweezer approximately 3 μm above the surface of the motility chamber. The cell is constrained by the glass surface. However, the rotation and out-of-plane beating is observable. Movie was recorded using wide-field microscopy with a 60x objective at 45 fps and played back at the same frame rate.
(AVI)

**S3 Movie. LC2 knockdown cells exhibiting t high directional persistence motility.** The movie was recorded using phase-contrast microscopy with a 10x objective at 45 fps and played

back at the same frame rate. The scale bar represents 10 μm.
(AVI)

**S4 Movie. LC2 knockdown cells exhibiting medium directional persistence motility.** The movie was recorded using phase-contrast microscopy with a 10x objective at 45 fps and played back at the same frame rate. The scale bar represents 10 μm.
(AVI)

**S5 Movie. LC2 knockdown cells exhibiting low directional persistence motility.** The movie was recorded using phase-contrast microscopy with a 10x objective at 45 fps and played back at the same frame rate. The scale bar represents 10 μm.
(AVI)

**S6 Movie. FLAM3-LC2 KD double knockdown cells show altered ciliary beating.** A high-speed movie of FLAM3-LC2 KD double knockdown cells showing a highly asymmetric forward (tip-to-base) beating with an intermittent reversal in ciliary beat direction—both of the phenomena leading to cell reorientation. The movie was recorded at 200 fps with an exposure time of 250 μs and played back at 45 fps.
(AVI)

**S7 Movie. FLAM3-LC2 KD double knockdown cells exhibit altered ciliary beating leading to frequent cell reorientation.** A typical freely swimming FLAM3-LC2 KD cell showing frequent reversals in ciliary beat mode and futile swimming motility with low directionality. The movie was recorded at 45 fps and played back at the same frame rate. The scale bar = 10 μm in both movies.
(AVI)

**S8 Movie. FLAM3-LC2 KD double knockdown cells exhibit ciliary beating locked to a reverse beating over an extensive period showing a longer dwell time in reverse beating mode.** The movie was recorded using a high-speed CMOS camera and a 60x objective at 200 fps with an exposure time of 250 μs and played back at 45 fps. Scale bar = 10 μm.
(AVI)

**S9 Movie. The trypanosome base-to-tip beating mode is not exclusively asymmetric.** Movie of a typical WT/LC2 OE cell exhibiting extensive, a largely symmetric, base-to-tip (reverse) beating mode. This shows the effect of LC2::eGFP overexpression on trypanosome cell motility. The movie was recorded using phase-contrast microscopy at 45 fps and played back at the same frame rate. Scale bar = 10 μm.
(AVI)

**S1 File. Homology modeled structure of TbLC2 using CrLC2 as a template generated by SWISS-MODEL and used and modified under the creative commons license CC BY-SA 4. 0.**
(PDB)

**S2 File. Template-free model of TbLC2 built using AI-based AlphaFold modeling with the 22 amino acid N-terminal truncation.**
(PDB)

**S3 File. Template-free model of the full-length TbLC2 built using AI-based AlphaFold modeling.**
(PDB)

## Acknowledgments

We are grateful to the Clemson Light Imaging Facility, Terri Bruce, and Rhonda Reigers Powell as well as the Clemson College of Science shared microscopy facility for the use of and their help using the light microscopes. We are also grateful to the Clemson Electron Microscopy Facility, Laxmikant Vyankatesh Saraf, and George Wetzel for the use of and their help using the TEM and SEMs.

We also acknowledge Aliona Bogdanova (Max Planck Institute for Molecular Cell Biology and Genetics), Jim Morris (Clemson University), members of the J. Morris Lab, Meredith Morris (Clemson University), Kent Hill (UCLA), and Ritsu Kamiya (Kyoto University) for various reagents and samples, use of their equipment, and general advice. We also thank Veikko Geyer and Benjamin Friedrich for the MATLAB source code we adapted to track beating trypanosome cilia. Finally, we thank Marija Zanic for her careful reading of the manuscript and many fruitful discussions.

## Author Contributions

**Conceptualization:** Joshua Daniel Alper.

**Data curation:** Subash Godar, Joshua Daniel Alper.

**Formal analysis:** Subash Godar, Joshua Daniel Alper.

**Funding acquisition:** Joshua Daniel Alper.

**Investigation:** Subash Godar, James Oristian, Valerie Hinsch, Katherine Wentworth, Ethan Lopez, Parastoo Amlashi, Gerald Enverso, Joshua Daniel Alper.

**Methodology:** Subash Godar, James Oristian, Valerie Hinsch, Katherine Wentworth, Ethan Lopez, Parastoo Amlashi, Gerald Enverso, Samantha Markley, Joshua Daniel Alper.

**Project administration:** Subash Godar, Joshua Daniel Alper.

**Resources:** Subash Godar, James Oristian, Valerie Hinsch, Katherine Wentworth, Gerald Enverso, Samantha Markley, Joshua Daniel Alper.

**Software:** Subash Godar, Joshua Daniel Alper.

**Supervision:** Joshua Daniel Alper.

**Validation:** Subash Godar, Joshua Daniel Alper.

**Visualization:** Subash Godar, Joshua Daniel Alper.

**Writing – original draft:** Subash Godar, Joshua Daniel Alper.

**Writing – review & editing:** Subash Godar, Joshua Daniel Alper.

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
